# Long noncoding RNA *SAM* promotes myoblast proliferation through stabilizing Sugt1 and facilitating kinetochore assembly

Yuying Li[1], Jie Yuan[1], Fengyuan Chen[2], Suyang Zhang[2], Yu Zhao[2], Xiaona Chen[2], Leina Lu[1,3,6], Liang Zhou[4], Ching Yan Chu[5], Hao Sun[1,3✉] & Huating Wang[2,3✉]

The functional study of lncRNAs in skeletal muscle satellite cells (SCs) remains at the infancy stage. Here we identify *SAM* (Sugt1 asssociated muscle) lncRNA that is enriched in the proliferating myoblasts. Global deletion of *SAM* has no overt effect on mice but impairs adult muscle regeneration following acute damage; it also exacerbates the chronic injury-induced dystrophic phenotype in mdx mice. Consistently, inducible deletion of *SAM* in SCs leads to deficiency in muscle regeneration. Further examination reveals that *SAM* loss results in a cell-autonomous defect in the proliferative expansion of myoblasts. Mechanistically, we find *SAM* interacts and stabilizes Sugt1, a co-chaperon protein key to kinetochore assembly during cell division. Loss of *SAM* or Sugt1 both disrupts kinetochore assembly in mitotic cells due to the mislocalization of two components: Dsn1 and Hec1. Altogether, our findings identify *SAM* as a regulator of SC proliferation through facilitating Sugt1 mediated kinetochore assembly during cell division.

[1] Department of Chemical Pathology, The Chinese University of Hong Kong, Hong Kong, China. [2] Department of Orthaepedics and Traumatology, The Chinese University of Hong Kong, Hong Kong, China. [3] Li Ka Shing Institute of Health Sciences, The Chinese University of Hong Kong, Hong Kong, China. [4] Department of Toxicology, School of Public Health, Southern Medical University, Guangzhou, China. [5] Department of Obstetrics and Gynaecology, Li Ka Shing Institute of Health Sciences, The Chinese University of Hong Kong, Hong Kong, China. [6] Present address: Department of Genetics and Genome Sciences, School of Medicine, Case Western Reserve University, Cleveland 44106 OH, USA. ✉email: haosun@cuhk.edu.hk; huating.wang@cuhk.edu.hk

Skeletal muscle has a robust regenerative capacity, which mainly relies on the activation of resident muscle stem cells, termed satellite cells (SCs). These cells are uniquely marked by the expression of paired box 7 (Pax7) protein and normally lie in a niche beneath the basal lamina of myofibers in a quiescent stage. Upon injury, they are rapidly activated to enter the cell cycle and undergo proliferative expansion as myoblast (MB) cells which then differentiate and fuse to form multinucleated myotube (MT) cells; these myotubes further mature into myofibers to restore the damaged muscle. Meanwhile, a subset of SCs exit the cell cycle and return to the quiescent stage for replenishing the adult stem cell pool. Fine-tuned regulation of cell cycle is thus essential to ensure appropriate progression through the various overlapping states: activation, proliferation, differentiation, and self-renewal/returning to quiescence.

The cell cycle involves DNA replication and subsequent chromosome separation. The faithful chromosome segregation relies on the assembly of mitotic kinetochore on centromeric chromatin to mediate its interaction with spindle microtubules[1]. In vertebrates, the kinetochore is a multilayered disc structure that contains more than a hundred of proteins components[2]. CCAN, the constitutive centromere-associated network, is restricted to the centromeres throughout the cell cycle forming a major component of inner kinetochore whereas KMN network, including the KNL1 complex (containing Knl1 (kinetochore scaffold 1), and Zwint (ZW10 interactor)), the MIS12 complex (containing Mis12, Dsn1, Pmf1 (polyamine-modulated factor 1) and Nsl1), and the NDC80 complex (containing Ndc80 (also called Hec1), Nuf2, Spc24, and Spc25), is recruited to the centromere by the CCAN during specific stages of mitosis, forming prominent subunits of outer kinetochore[3,4]. Among the KMN complexes, Mis12 complex is the keystone to serve as a protein interaction hub which assembles outer kinetochore and links to the inner kinetochore[5]. Ndc80 complex directly interacts with microtubules through its component Hec1[6]. Given the large number of kinetochore components, its proper assembly is a dynamic and highly orchestrated process. Any error in kinetochore assembly such as improper targeting or turnover of any component can affect the progression of mitosis, leading to disrupted microtubule attachment, improper chromosomal segregation, the formation of multipolar spindles, mitotic delay or aneuploidy, etc.[7–9]. It is thus important to elucidate the regulatory mechanisms facilitating kinetochore assembly, which has not been done in SCs. It is known that SGT1, suppressor of G2 allele of SKP1 (*S. cerevisiae*) (Sugt1) is a highly conserved protein involved in kinetochore assembly[10]. As a co-chaperone for Hsp90 protein, mammalian Sugt1 ensures efficient formation of microtubule-binding sites by recruiting Mis12 complexes to kinetochore[11]. Reduction of Sugt1 in Hela cells leads to destabilization and mis-localization of Dsn1 and Hec1, thus causing inefficient formation of high-affinity kinetochore-microtubule attachment sites and a mitotic delay[10,11]. A recent study also showed that a regulatory phosphatase PHLPP1 dephosphorylates Sugt1 thereby prevents Sugt1 from associating with E3 ligase in turn, countering Sugt1 ubiquitination and degradation during kinetochore formation[12].

Long non-coding RNAs (lncRNAs) are emerging as a family of gene regulators of skeletal muscle regeneration and SC activities. Thousands of lncRNAs have been identified in skeletal muscle cells but our understanding of lncRNA participation in skeletal myogenesis is still at the infancy stage with only a handful of reports from our group and others[13–18]. Most efforts concentrated on illuminating their regulatory mechanisms in the transition of MB into MT using a mouse MB line, C2C12[13–15]; it remains largely uncharacterized whether lncRNAs can regulate other states of SCs. In terms of underpinning molecular mechanisms, lncRNAs are best known for engaging in transcriptional and epigenetic regulation on chromatins, usually through their interaction with chromatin regulators[19]; other unique mechanisms are also being uncovered to explain the diversified modes of lncRNA actions. For example, recently, lncRNAs generated from the repeat region of centromere in Drosophila and human, were found to bind to the kinetochore component CENP-C, adding lncRNA to the complex epigenetic marks at centromeres[20,21]. Still, it is not known whether non-centromeric lncRNAs exist to interact with proteins involved in kinetochore assembly. Additionally, in vivo functional analysis is in general lacking for most lncRNAs studied so far despite a wealth of knowledge accumulated from using in vitro cell culture; to date there have been only a few lncRNA genetically studied using knockout (KO) animals[22,23].

Here, we have identified one lncRNA, *SAM*, as a regulator of MB proliferation. Its expression is evidently upregulated when SCs undergo active proliferation; knockdown of *SAM* in vitro delays proliferative expansion of cells. To further investigate its function in vivo, we generated a KO mouse of *SAM* using KO-first strategy; loss of *SAM* does not cause overt phenotype but indeed leads to impaired regeneration after acute injury. Consistently, inducible deletion of *SAM* in SCs also delays the process of acute injury-induced muscle regeneration. Moreover, deletion of *SAM* in a dystrophic mdx mouse exacerbates the chronic injury-induced dystrophic phenotype. Further examination reveals that *SAM* deletion results in the cell-autonomous defect in MB proliferation, pointing to *SAM* as a promoting factor of MB proliferation. High throughput identification of *SAM* interacting protein partners reveals that it can specifically bind to Sugt1 and stabilizes its protein level in MBs; loss of *SAM* causes increased ubiquitination of Sugt1. Mechanistically, *SAM* facilitates Sugt1-mediated kinetochore assembly. Loss of *SAM* or Sugt1 both causes disrupted chromosome alignment and microtubule attachment, which is likely a result of mis-localization of Dsn1 and Hec1 proteins in centromere. Altogether our findings have identified *SAM* as a regulator of MB proliferation through its synergistic action with Sugt1 to promote kinetochore assembly during cell division.

## Results

**SAM is enriched in MB and promotes cell proliferation.** Previously we have defined dozens of uncharacterized lncRNAs from C2C12 MB vs. MT cells through de novo discovery approach integrating RNA-seq and ChIP-seq datasets[13]. One lncRNA, *Gm11974*, named as Sugt1 Associated Muscle (*SAM*) lncRNA in the present study, displayed relatively high expression and unexplored function in MB cells (Fig. 1a). It localizes on mouse chromosome 11, in the intervening region of *Myo1g* (Myosin IG) and *Ccm2* (Cerebral cavernous malformation 2) protein-coding genes (Fig. 1b), with well-defined gene structure and a binding peak of myogenic master transcription factor, MyoD on its promoter region (Fig. 1a). A human homolog of this gene, *SNHG15*, has been studied in cancer, showing upregulated expression in multiple tumor tissues or cells[24–26] and it promotes cancer cell proliferation and migration by serving as a sponge for miR-NAs[27–29]. Through rapid amplification with cDNA ends (RACE), one dominant isoform was cloned from C2C12 MB cells, which was 592 bp long with four exons (Fig. 1c). It was predicted as a non-coding RNA by iSeeRNA[30] (Supplementary Fig. 1a), consistent with its annotation in the RefSeq (Accession no. NR 045893). *SAM* was readily detected in C2C12 MBs and down-regulated when the cells underwent differentiation to form MTs (Supplementary Fig. 1b). Consistently, it was enriched in the

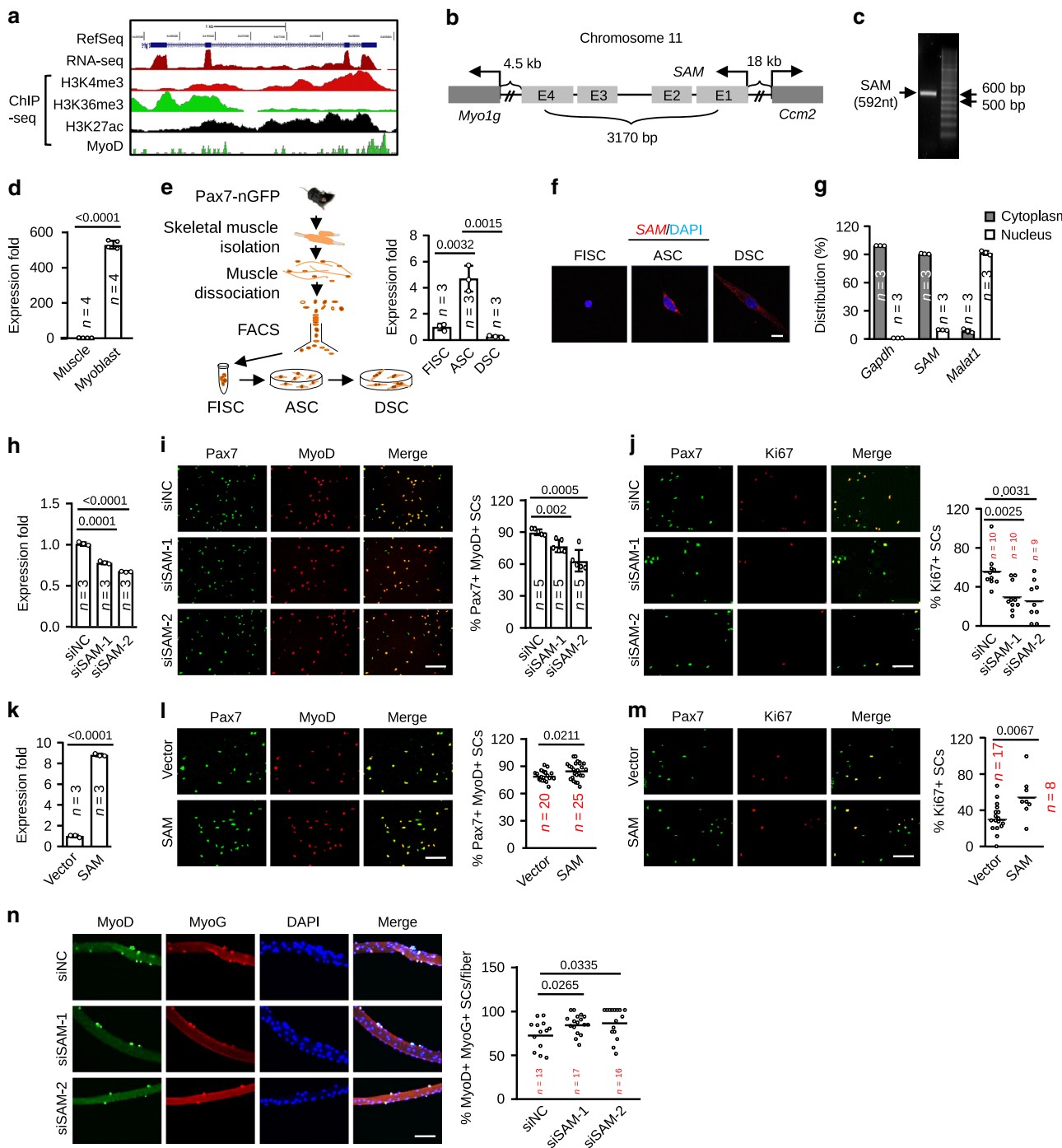

**Fig. 1 SAM is enriched in myoblast and promotes cell proliferation in vitro. a** Genomic snapshot of mouse *SAM* generated in RefSeq, RNA-seq, histone marks ChIP-seq from activated satellite cells (ASCs), MyoD ChIP-seq from C2C12 myoblasts. **b** Schematic illustration of the genomic location and structure of mouse *SAM* (*Gm11974*) locus. **c** Semi-quantitative RT-PCR detection of full-length *SAM* transcript (592 nt) in C2C12 myoblasts. **d** qRT-PCR analysis of *SAM* in mouse muscle vs. isolated primary myoblasts. **e** qRT-PCR detection of *SAM* in freshly isolated SCs (FISCs), activated SCs (ASCs), or differentiated SCs (DSCs) isolated from muscles of Tg: Pax7-nGFP mice. **f** FISH was performed in the above FISCs, ASCs, or DSCs. **g** qRT-PCR analysis of RNAs purified from cytoplasmic or nuclear fraction of ASCs. **h** qRT-PCR detection of *SAM* from ASCs transfected for 48 h with either control (siNC) or *SAM* siRNA (si*SAM*-1 or si*SAM*-2). **i** Immunofluorescence (IF) staining for Pax7 and MyoD or **j** Ki67 was performed in the above transfected cells and the percentage of positively stained cells was quantified. **k** qRT-PCR detection of *SAM* from ASCs transfected for 48 h with a Vector or *SAM* expressing plasmid. **l** IF staining for Pax7 and MyoD or **m** Ki67 was performed in the above transfected cells. **n** Single myofibers were isolated from EDL muscles of adult C57BL/6 mouse and transfected with si*SAM* oligos. IF staining of MyoD and MyoG was performed 72 h after transfection and the percentage of positively stained cells was quantified. The data are presented as mean ± SD in **d**, **e**, **g**–**i** and **k**. The center line in **j**, **l**, **m**, and **n** is presented as mean. The *p* values by two-tailed unpaired *t* test are indicated in **d**, **e**, **h**–**n**. The total number of biologically independent samples are indicated in **d**, **e**, **g**–**n**. Scale bars: 10 μm **f**, 100 μm **i**, **j**, **l**, **m**, and **n**. Source data are provided as a Source Data file.

primary MBs isolated from the skeletal muscle compared with the whole muscle tissue (Fig. 1d). To further examine its expression dynamics during SC lineage progression, freshly isolated SCs (FISCs) from limb muscles of Pax7-nGFP mice[31] were cultured with growth medium to become activated (ASCs or MBs) which were further cultured to differentiate (DSCs); *SAM* level was evidently induced (4.7 fold) in ASCs vs. FISCs but decreased sharply (72.71%) in DSCs vs. FISCs (Fig. 1e). Interestingly, *SAM* expression appeared not to be heterogeneous in SCs, since no significant difference was detected in the isolated Pax7$^{High}$ and Pax7$^{Low}$ subpopulations[32] of FISCs or ASCs (Supplementary Fig. 1c, d). The above results suggested that *SAM* might promote MB proliferation. RNA fluorescence in situ hybridization (RNA-FISH) analysis revealed that *SAM* transcripts mainly distributed in the cytoplasm of SC (Fig. 1f); a stronger signal was detected in ASC vs. FISC or DSC. Similarly, the predominant cytoplasmic localization was also observed in C2C12 MB but decreased in MT (Supplementary Fig. 1e). Consistently, cellular fractionation assay in ASCs (Fig. 1g) or C2C12 (Supplementary Fig. 1f) also showed that *SAM* transcripts were enriched in cytoplasmic extracts, in a similar pattern as *Gapdh* transcripts, whereas lncRNA *Malat1* was only found in nuclear extracts[16]. The unique cytoplasmic localization of *SAM* suggested that its function may be distinct from many lncRNAs that are involved in transcriptional regulation in myogenesis[17], which therefore triggered our further investigation.

To test if *SAM* is required for efficient MB proliferation, we knocked down *SAM* expression in ASCs with two different siRNA oligos (22.98% and 33.27% reduction, respectively) (Fig. 1h). Forty-eight-hour post-transfection, SCs were stained for Pax7 and MyoD to evaluate the degree of proliferation. It is known that fully activated SCs are marked by both Pax7 and MyoD while self-renewing SCs express Pax7, but not MyoD; In DSCs, Pax7 expression is lost while Myogenin (MyoG) expression increases[33]. Indeed, the percentage of Pax7+MyoD + cells was markedly reduced upon siSAM knockdown (14.43% and 29.85%, respectively) (Fig. 1i, Supplementary Fig. 1g and h). This was further confirmed by staining for Ki67; the percentage of Ki67+ SCs was decreased upon *SAM* loss (48.11% and 55.25%) (Fig. 1j). On the contrary, when overexpressing *SAM* by transfecting a *SAM*-expressing plasmid (Fig. 1k), an increase in the percentage of Pax7+MyoD+ (7.16%) or Ki67+ (74.89%) cells was observed (Fig. 1l, m, Supplementary Fig. 1i and j). Altogether, the above results from loss and gain-of-function assays in vitro on SCs demonstrated that *SAM* promotes MB proliferation. When performing similar assays using C2C12 MB cell line with stable *SAM* knockdown by a shRNA, the same overall conclusions were reached (Supplementary Fig. 1k–p). In addition, by cell cycle analysis of synchronized cells, sh*SAM* cells displayed a higher percentage of cells in G1 phase at both 12 h and 24 h compared with control (Ctrl) cells, suggesting *SAM* loss caused cell cycle arrest at G1 phase (Supplementary Fig. 1q); nevertheless, *SAM* expression did not show dynamic pattern during the cell cycle progression (Supplementary Fig. 1r). Next, we also examined whether the loss of *SAM* has any effect on MB differentiation. Single myofibers were isolated from extensor digitorum longus (EDL) muscle of mouse and transfected with si*SAM*. Staining with MyoG and MyoD 72 h post-transfection revealed that the percentage of MyoD+MyoG+ cells was significantly increased (16.43% and 19.53%) upon *SAM* knockdown (Fig. 1n, Supplementary Fig. 1s and t), implying that these cells may have precocious differentiation potential. Collectively, these findings from the in vitro cultured cells indicate that *SAM* is necessary for maintaining proper myogenic proliferation and preventing precocious differentiation.

**SAM deletion in mouse impairs muscle regeneration.** To further elucidate the functional roles of *SAM* in vivo, we generated a KO mouse. Given that lncRNA locus may function as an enhancer region to regulate gene expression[34] and active enhancer mark, H3K27ac, was indeed found on *SAM* locus (Fig. 1a), we thus employed a KO-first strategy that ablates gene function by inserting RNA processing signals without deletion of the target locus. As illustrated in Fig. 2a, the KO-first allele was generated by inserting a splicer acceptor (SA)-internal ribosomal entry site (IRES)-LacZ cassette and a Neo-polyadenylation (pA) signal into the intron 2, thus achieving the disruption of *SAM* transcription. The insertion was flanked by FRT sites that will allow Flippase recombinase to remove the gene-trapping cassette, hereby converting the KO to a conditional allele with loxP sites flanking exons 3–4. DNA genotyping confirmed the insertion of the SA-IRES-LacZ-pA cassette in the KO mouse genome (Supplementary Fig. 2a, b). Three qRT-PCR primers targeting different regions (exons 1–2, exons 2–3, and exons 3–4) were used to detect possible transcription (Fig. 2a); and no transcripts were detected with any pair of primers in the isolated SCs (Fig. 2b) or tested tissues (Supplementary Fig. 2c). It is interesting that no truncated transcript from exons 1–2 was detected despite the PolyA was inserted after exon 2. To test if non-sense-mediated RNA decay had possibly led to degradation of the transcript, we found treatment with cycloheximide (CHX), which is known to reverse non-sense-mediated RNA decay[35] induced the appearance of the truncated transcript from exons 1–2 (Supplementary Fig. 2d). Examining the adult mouse phenotype, we found that the KO mice were viable, fertile without overt morphological deformities (Fig. 2c); consistently, the size and weight of the KO mice were comparable with the WT littermates (Fig. 2c, d). Histological analyses of adult tissues including liver, spleen, lung, kidney, and ovary also revealed no overt differences between the KO and WT littermates (Supplementary Fig. 2e); similarly, when examining the adult skeletal muscle at 8 weeks, the fibers also appeared normal in size and pattern (Fig. 2e), showing the deletion of *SAM* may not have any impact on the adult muscle development. In addition, the number of Pax7+ quiescent SCs (QSCs) was not changed in the muscles of the KO mice (Fig. 2f), indicating that *SAM* may not be required for maintenance of the SC pool. Lastly, examining muscle formation at embryonic (E18.5) (Supplementary Fig. 2f, g) or postnatal (P7) days (Supplementary Fig. 2h) revealed no overt changes in muscle morphology and the number of Pax7+ cells in WT vs. KO mice, suggesting that *SAM* may not play a role in embryonic or postnatal myogenesis.

Considering the promoting function of *SAM* that was uncovered above in proliferating MB in vitro (Fig. 1), we speculated that loss of *SAM* may have an impact on muscle regeneration in vivo. To test this notion, $BaCl_2$ was injected into the tibialis anterior (TA) muscles of 8–9 weeks old mice to induce massive myofiber necrosis followed by immune cell infiltration, activation, and proliferation of SCs, which then formed new myofibers to repair the damaged fibers within 3–4 weeks post injection. The newly formed myofibers were normally characterized by centrally localized nuclei (CLN) and expression of embryonic MyHC (eMyHC) protein. The above injected muscles were collected 4, 7, 14, and 28 days after the injury for evaluation of the degree of muscle regeneration (Fig. 2g). Indeed, by H&E staining, the number of CLN+ fibers per field was evidently decreased (32.29%) in KO vs. WT mice 4 days after the injury (Fig. 2h); consistently, the number of eMyHC+ fibers was also decreased by 20.16% (Fig. 2i). Nevertheless, by day 7, no significant difference was found in KO vs. WT mice; by day 28, the damaged muscle fibers were fully regenerated in both mice (Fig. 2h, i). The above results indicate that *SAM* deletion causes a delay but not a blockage in injury-induced muscle regeneration.

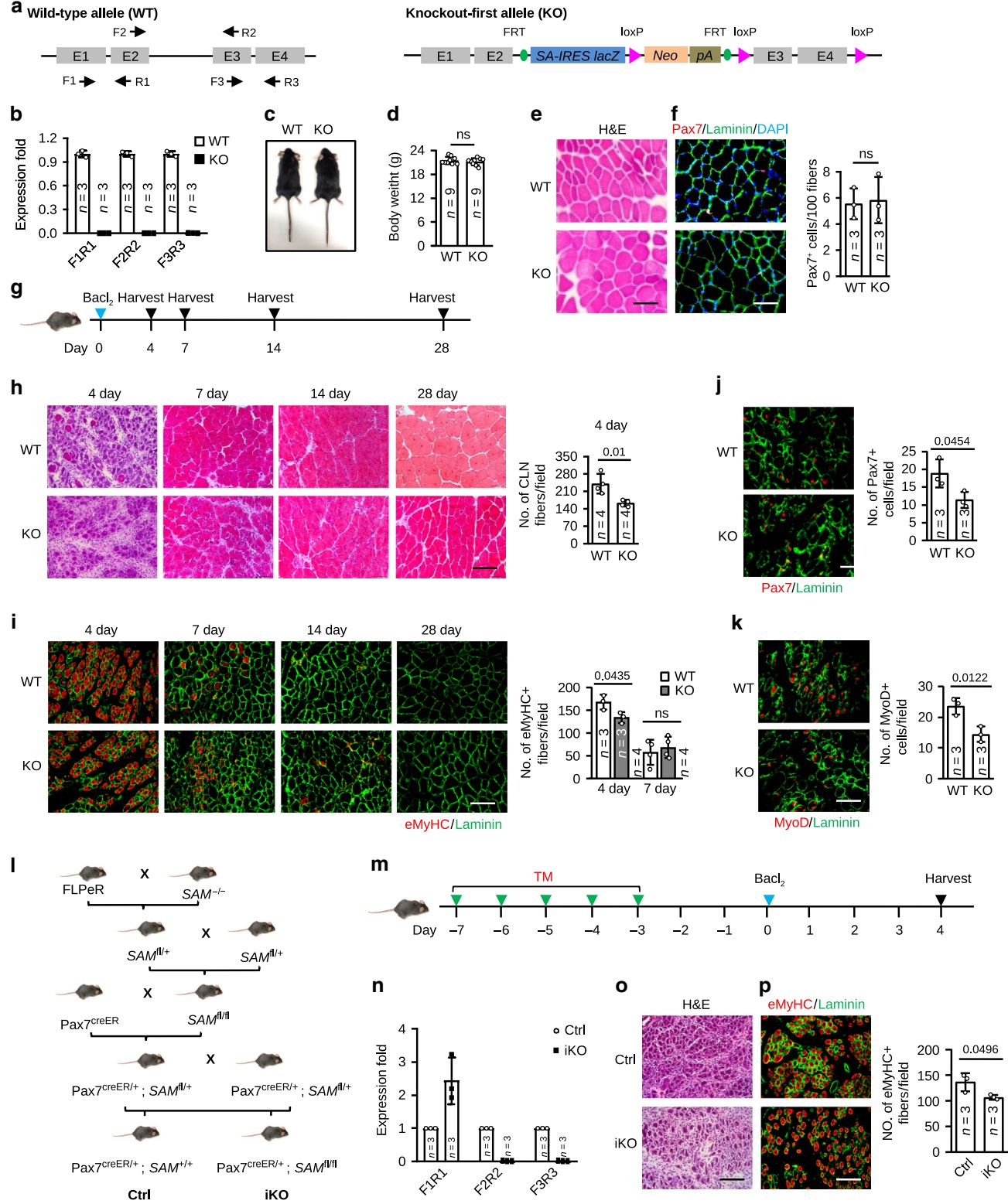

In addition, we found that Pax7+ or MyoD+ cells were both reduced significantly (39.44% and 38.81%, respectively) on the KO muscles compared with WT (Fig. 2j, k) 3 days after injury, suggesting a decline in the expansion of SC progeny during the regeneration process. Lastly, we quantified the number of Pax7+ cells one month after injury when SCs were expected to return to quiescence (Supplementary Fig. 2i); no significant difference was observed from the injured muscles of KO vs. WT

mice (Supplementary Fig. 2j), suggesting *SAM* ablation may not exert apparent influence on SC self-renewal during muscle regeneration.

**Inducible ablation of *SAM* in SC delays muscle regeneration.** To further pinpoint that the above described regeneration phenotype is attributed to the loss of *SAM* in SCs, we further generated a SC-specific inducible knockout (iKO) mouse. As

**Fig. 2 Constitutive or inducible *SAM* deletion impairs muscle regeneration. a** Schematic illustration of wild-type (WT) and *SAM* knockout-first (KO) mouse alleles. SA splice acceptor, IRES internal ribosome entry site, Neo neomycin, pA polyadenylation signal. Arrows indicate the locations of primers used for qRT-PCR. **b** qRT-PCR detection of *SAM* in FISCs. **c** Representative images of adult WT and KO mice. **d** Body weight from age-matched WT and KO mice. **e** Hematoxylin and Eosin (H&E) staining of tibialis anterior (TA) muscle from WT and KO mice. **f** IF staining for Pax7 and Laminin on the above muscles. Pax7+ SCs per 100 fibers were quantified. **g** The scheme for $BaCl_2$ injection induced TA muscle injury and subsequent analyses. **h** H&E staining was performed on the above muscles. The centrally localized nuclei (CLN) fibers were quantified at day 4 post-injury. **i** eMyHC and Laminin immunostaining was performed in **g** harvested muscles. eMyHC+ fibers were quantified at day 4 and 7 post-injury. **j** IF staining for Pax7 and Laminin or **k** MyoD and Laminin was performed in the TA muscles 3 days post-injury. Positively stained cells were quantified. **l** Breeding scheme for generating Control (Ctrl) and inducible *SAM* knockout mice (iKO). **m** Schematic of Tamoxifen (TM) injection, $BaCl_2$ injection, and SC collection in Ctrl or iKO mice. **n** qRT-PCR detection of *SAM* in FISCs 3 days after TM injection. **o** H&E staining was performed in the TA muscles 4 days after injury. **p** IF staining for eMyHC and Laminin in the above muscles was performed and eMyHC+ fibers were quantified. The data are presented as mean ± SD in **b**, **d**, **f**, **h–k**, **n**, and **p**. The *p* values by two-tailed unpaired *t* test are used for comparing two groups, ns not significant. The total number of mice used are indicated in **b**, **d**, **f**, **h–k**, **n**, and **p**. Scale bars: 50 μm (**e**, **f**, **j**, and **k**), 100 μm (**h**, **i**, **o**, and **p**). Source data are provided as a Source Data file.

illustrated in Fig. 2l, *SAM* floxed mice ($SAM^{fl/fl}$) were created by crossing the KO with a FLPeR recombinase mouse, which led to the excision of the SA-IRES-LacZ-pA cassette flanked by FRT sites (Supplementary Fig. 2k, l). Further breeding with a Pax7$^{creER}$ mouse[36] to generate Pax7$^{creER/+}$; $SAM^{fl/fl}$ mouse (termed *SAM* iKO) led to permanent deletion of exons 3–4 of *SAM* in the adult Pax7+ SCs following five consecutive doses of tamoxifen (TM) injection in 2-month-old mouse (Fig. 2m); the successful elimination of exons 3–4 of *SAM* was confirmed (Fig. 2n); interestingly, a truncated transcript was generated from exons 1–2 (Fig. 2n). Consistent with what was observed in the KO mouse (Fig. 2h, i), the iKO mouse also displayed impaired regenerative ability after $BaCl_2$ induced muscle injury as assessed by a 21.95% decreased number of eMyHC+ 4 days after injury (Fig. 2o, p). Taken together, findings from both KO and iKO mice solidify our thinking that *SAM* is necessary for the timely repair of damaged skeletal muscle tissue after acute injury.

**SAM deletion aggravates dystrophic phenotype in mdx mouse.**
Besides acute injury by $BaCl_2$ injection, innate genetic defects can also provoke chronic injury-induced muscle regeneration. For example, in the widely used mouse model for Duchenne muscular dystrophy (DMD), mdx mouse displays extensive muscle degeneration and regeneration as early as ~3 weeks of age; repetitive degeneration/regeneration cycles lead to the eventual loss of SC regenerative capacity and fatty fibrosis in old mdx mouse[37,38]. To examine whether *SAM* loss may affect chronic injury-induced regeneration in DMD, we generated *SAM*; dystrophin double KO (dKO) mouse by crossing the *SAM* KO first mouse with mdx mouse (Fig. 3a). As expected, *SAM* expression was completely depleted in freshly sorted SCs of dKO vs. control (Ctrl) mdx mice (Fig. 3b). The dKO mouse displayed no overt difference from the Ctrl mouse (Fig. 3c) with comparable body weight during the course of 27 weeks (Supplementary Fig. 3a); TA and gastrocnemius (GAS) muscles also showed comparable weight at 8 weeks (Supplementary Fig. 3b). However, when examined closely, smaller myofibers were more frequently observed in the TA muscles of 8 weeks old dKO mouse as measured by the cross-sectional area (CSA) of individual fiber (Fig. 3d). Moreover, histological examination revealed increased size of unrepaired areas (Fig. 3e), an increased number of eMyHC+ myofibers (Fig. 3f) and increased infiltration of CD68+ macrophages (Fig. 3g) in dKO mice, suggesting loss of *SAM* delays the muscle regeneration in limb muscles. Compared to limb muscles, mdx diaphragm (Dia) muscle is known to exhibit a more severe dystrophic phenotype manifested by fibrosis and fatty infiltration that worsens as mice age[33,39]. Expectedly, the dKO mice at 6–8 months displayed the exacerbation of fibrosis as evidenced by increased Collagen I or Trichrome staining

(Fig. 3h–j). Taken together, the above results suggest that loss of *SAM* aggravates dystrophic phenotype of mdx mice.

**Loss of SAM leads to SC autonomous defects in proliferation.**
To further elucidate the impact of *SAM* loss on SC activities, we tested whether *SAM* loss impaires MB proliferation in the KO mouse (Fig. 4). First, in vivo EdU labeling after $BaCl_2$ injury indeed revealed a reduced percentage (12.33%) of proliferating MBs in KO vs. WT littermates (Fig. 4a and Supplementary Fig. 4a); similar reduction (14.67%) was also observed when the assay was performed in iKO vs. Ctrl littermates (Fig. 4b). To further elucidate whether this proliferative defect is cell-autonomous, FISCs from KO or WT mice were cultured for 2 days and a lower percentage of EdU+ cells in KO (57.01%) vs. WT (66.91%) cells was detected (Fig. 4c). Furthermore, a significant reduction of the percentage of Pax7+MyoD+ cells was observed in KO (85.61%) vs. WT (91.78%) cells, suggesting a decline in the proliferative capacity of MBs (Fig. 4d). Consistently, when performed on SCs isolated from iKO mouse, the same conclusion was reached; a reduced percentage of EdU+ (25.61%) or Pax7+MyoD+ cells (5.59%) was found in iKO vs. Ctrl cells (Fig. 4e, f). Moreover, we also isolated single myofibers and performed the above assays in SCs associated with the cultured myofibers. Again, the percentage of EdU+ or Pax7+MyoD+ cells was significantly reduced in KO (10.88% and 5.35%, respectively) vs. WT cells (Fig. 4g and Supplementary Fig. 4b). In addition, MTS assay also revealed that SCs from KO muscle displayed a declining proliferating rate compared with WT control (Supplementary Fig. 4c). Of note, the impaired proliferation in KO SCs was rescued by re-expressing a *SAM* plasmid (Supplementary Fig. 4d, e and Fig. 4h), pinpointing loss of *SAM* as the cause of the deficient proliferation.

The above findings supported *SAM* loss inhibits proliferation in MBs. To further investigate if it also has any impact on other aspects of SC activities. We first found that within 30 h after isolation, the percentages of EdU+ and Pax7+ MyoD+ cells were reduced 36.62% and 17.91%, respectively in KO vs. WT (Fig. 4i, j), indicating a possible defect at the very early activation stage. Further assessing the differentiation ability, we found the percentage of MyoG+MyoD+ cells was increased (25.72%) in KO vs. WT SCs cultured for 3 days or myofiber-associated SCs cultured for the same period (19.87%) (Fig. 4k and Supplementary Fig. 4f). This was further substantiated by measuring the fusion index by MF20 staining after 2 days in DM; KO cells showed a higher fusion ability (2.7 fold) than WT cells (Fig. 4l), indicating *SAM* loss leads to an increased propensity for differentiation, which was consistent with the finding from Fig. 1n. Lastly, the TUNEL assay did not detect differences in KO vs. WT (Supplementary Fig. 4g) cells cultured for 2 days, suggesting *SAM* loss may not have caused SC apoptosis.

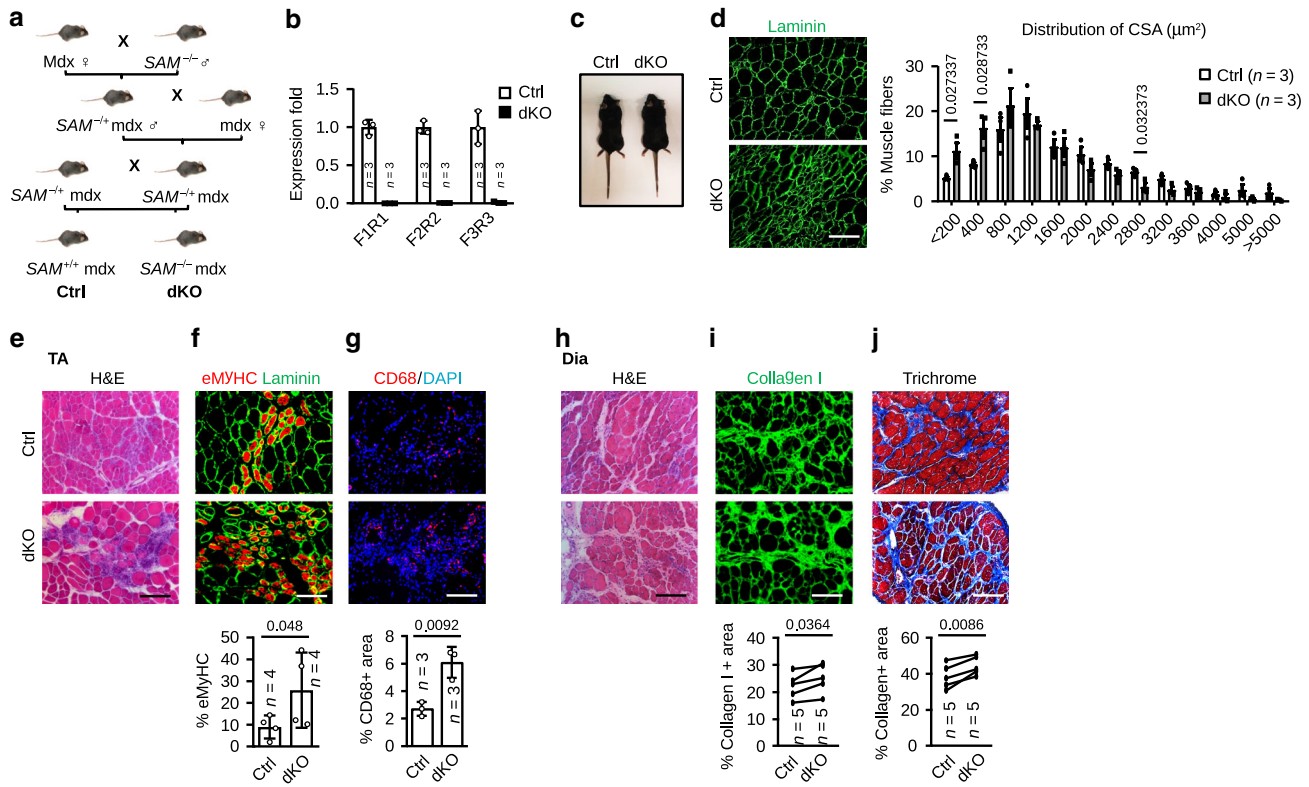

**Fig. 3 SAM deletion aggravates dystrophic phenotype in mdx mice. a** Schematic showing the generation of *SAM* and DMD double KO mice (dKO) through breeding mdx and *SAM* KO mice. **b** qRT-PCR was performed to confirm the loss of *SAM* in SCs of dKO vs. Ctrl mice. **c** Representative images of Ctrl and dKO littermates. **d** IF staining of Laminin was performed on TA muscles from 8 weeks old Ctrl and dKO mice and the cross-sectional area (CSA) of individual myofibers was quantified. The percentage of myofibers with a defined range of CSA over the total myofibers was calculated for each mouse. More than 1500 myofibers from three pairs of littermates were counted. **e** H&E and **f** eMyHC and Laminin staining was performed on the above TA muscle and the percentage of eMyHC+ fibers per field was quantified. **g** IF for CD68 was conducted on the above muscles and macrophage infiltration was assessed by quantifying the percentage of CD68 positive areas per field. **h** Diaphragm muscles (Dia)were isolated from 6 to 8 months old Ctrl and dKO mice and H&E was performed. **i** Staining of Collagen I was performed on the above Dia muscles and quantifications of the percentage of Collagen I positive areas per field are shown below the images. **j** Masson's Trichrome staining was performed on the above Dia muscles and the positively stained areas were quantified. The data are presented as mean ± SD in **b**, **f**, and **g** and mean ± SEM in **d**. The *p* values by two-tailed unpaired *t* test are indicated in **d** and **g**, two-tailed ratio paired *t* test in **f** and two-tailed paired *t* test in **i** and **j**. The total number of mice used are indicated in **b**, **d**, **f**, **g**, **i**, and **j**. Source data are provided as a Source Data file. Scale bars: 50 μm **j**, 100 μm **d**–**i**.

Altogether, the above results demonstrate that *SAM* deletion causes a delay in SC activation and proliferation but increases the propensity for precocious differentiation.

Lastly, the above phenotypical changes in cells were also substantiated when RNA-seq was performed to assess transcriptomic changes caused by *SAM* loss. The knock-down of *SAM* led to 250 genes down-regulated and 167 genes up-regulated in MBs (Supplementary Fig. 4h). Gene ontology (GO) cluster analysis revealed that the down-regulated genes were enriched for GO terms including cell cycle, M phase, microtubule-based process, chromatin assembly, etc. (Supplementary Fig. 4i), in line with the above uncovered function of *SAM* in promoting cell proliferation. The up-regulated genes were, on the other hand, enriched for skeletal system development, muscle cell differentiation, etc. (Supplementary Fig. 4j), which was consistent with the precocious differentiation phenotype upon *SAM* loss.

**SAM interacts with Sugt1 in MBs**. To dissect the molecular mechanism underlying *SAM* function in MBs, we sought to identify the interacting protein partners of *SAM* considering the well-known protein-binding ability of lncRNAs that endows themselves with many regulatory capacities[40]. To this end, we conducted RNA-pull down assay followed by mass spectrometry (MS) in C2C12 MBs using in vitro transcribed biotin-labeled

*SAM* or *GFP* transcripts[13] (Fig. 5a, b). A list of proteins was identified as potential interacting partners of *SAM*, among which Sugt1 caught our attention because of its known function in kinetochore assembly and cell mitosis[10,41]. Next, we confirmed the *SAM*/Sugt1 association by Western blotting following RNA pull-down. Indeed, an evident amount of Sugt1 was captured by *SAM*, but not *GFP* transcripts (Fig. 5c). No interaction was detected between *SAM* and a few other known RNA-binding proteins, Hnrnpl[42], Dnmt3a, and Dnmt3b[15], suggesting the specificity of the *SAM*/Sugt1 association. To further confirm their interaction, native RNA immunoprecipitation (RIP) assay was performed using an antibody against Sugt1 (Fig. 5d). A higher level (3.1 fold) of *SAM* was pulled down by the Sugt1 antibody vs. IgG control (Fig. 5d) while several control transcripts including *Gapdh*, *β-Actin* mRNAs, and lncRNA *Dum*[15] were not retrieved. Consistently, the co-labeling of *SAM* by RNA-FISH and Sugt1 protein by immunofluorescence (IF) revealed an evident co-localization of *SAM* with Flag-labeled Sugt1 in MBs (Fig. 5e). Altogether the above results substantiated that *SAM* specifically interacts with Sugt1 protein in MBs. In addition, we generated a series of deletion fragments of *SAM*, F1–F5 according to the predicted secondary structure by RNAfold (Supplementary Fig. 5a) and performed RNA-pulldown assay (Supplementary Fig. 5b) to map the binding domain of *SAM* with Sugt1.

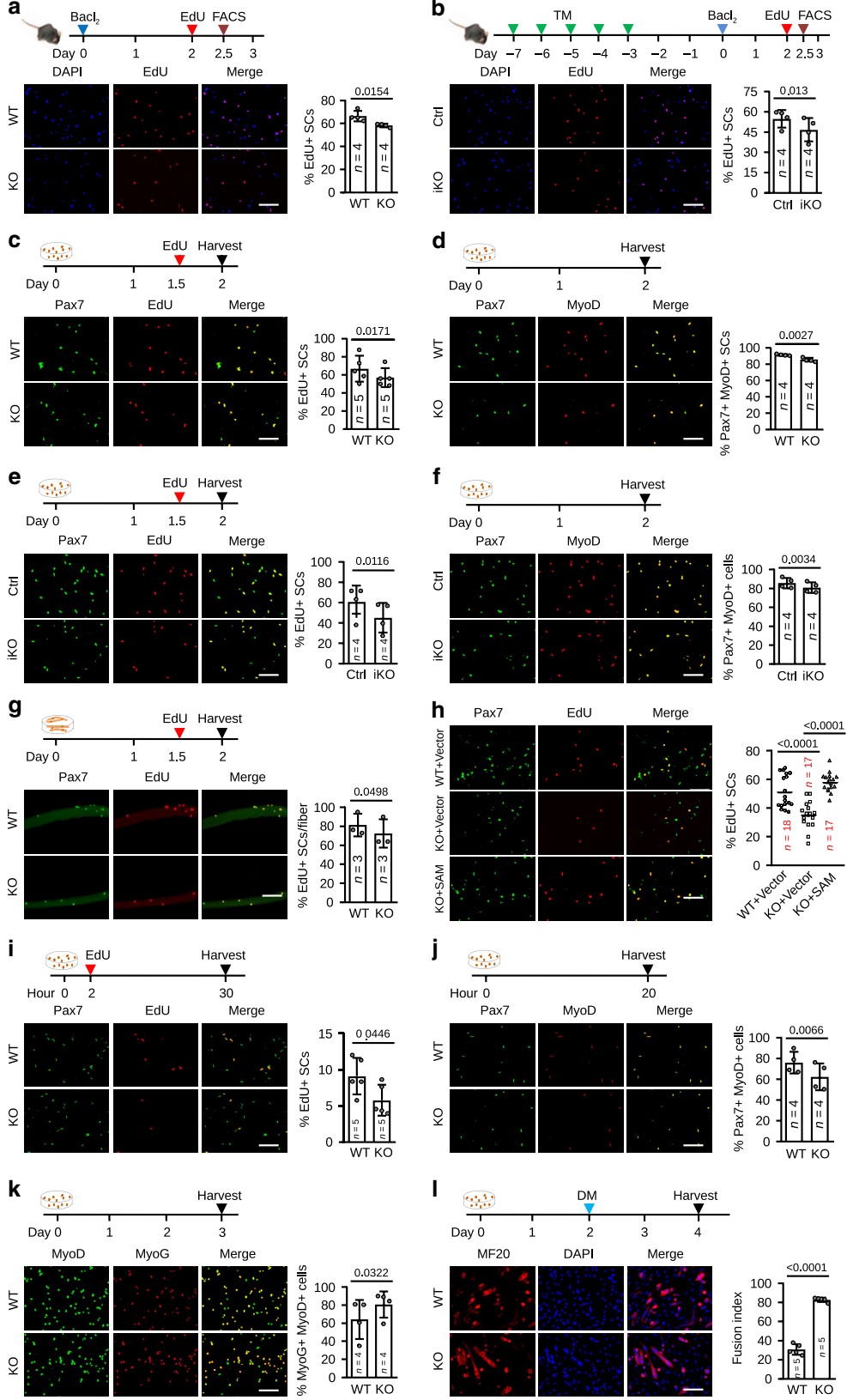

Interestingly, both F1 and F5 fragments of *SAM* retrieved comparable amounts of Sugt1 with the full-length transcripts. Nonetheless, the truncated transcript of exons 1–2 (containing F1 and F2) did not seem to be functional in muscle regeneration (Fig. 2n–p).

To further understand how Sugt1 and *SAM* together partake in the regulation of ASC proliferation, we found that similar to *SAM*, Sugt1 expression was also up-regulated upon SC activation at 24 h but down-regulated in differentiated cells at 96 h (Fig. 5f). Functionally, when knocked down Sugt1 in ASCs by two different

**Fig. 4 SAM loss in SCs leads to cell-autonomous defects in proliferation. a** Top: the experimental scheme for in vivo EdU assay in WT and KO or **b** Ctrl and iKO mice. Bottom: The percentage of EdU+ SCs was quantified. **c** Top: the experimental scheme for in vitro EdU assay in 48 h-cultured SC isolated from WT and KO mice. Bottom: The percentage of EdU+ cells was quantified. **d** The above cultured cells were stained for Pax7 and MyoD. The percentage of double-positive cells was quantified. **e** and **f** The above assays were performed in SCs isolated from Ctrl and iKO mice. **g** EdU assay was performed on single myofibers isolated from WT and KO mice. The percentage of EdU+ SCs was quantified. **h** EdU assay in ASC transfected with a Vector or SAM expressing plasmid. The percentage of EdU+ SCs was quantified. The center line is represented as mean. **i** EdU assay in 30 h-cultured SC isolated from WT and KO mice. The percentage of EdU+ SCs was quantified. **j** Pax7 and MyoD staining in 20h-cultured SCs isolated from WT and KO mice. The percentage of double-positive cells was quantified. **k** MyoD and MyoG staining in 3 days-cultured SCs isolated from WT and KO mice. Quantification of the double-positive cells was performed. **l** MF20 staining in 4 days-cultured SCs isolated from WT and KO mice. The fusion index of myotubes (≥2 nuclei)/total MF20+ cells) was quantified. DM differentiation medium. The data are presented as mean ± SD in **a–g** and **i–l**. The p values by two-tailed unpaired t test are indicated in **a**, **d**, **h**, and **l** and by two-tailed paired t test are indicated in **b**, **c**, **e**, **f**, **g**, **i**, **j**, and **k**. The total number of mice used in **a–g**, **i–k** and biologically independent samples in **h** and **l** are indicated. Scale bars: 100 μm **a–l**. Source data are provided as a Source Data file.

siRNA oligos (Fig. 5g), the proliferative ability of ASCs was reduced as shown by a decreased percentage of EdU+ cells (16.53% and 13.82%) compared to controls (Fig. 5h), phenocopying the effect of SAM loss. Moreover, the overexpression of Sugt1 (Supplementary Fig. 5c) fully rescued the deficient proliferation of the KO ASCs (Fig. 5i). Altogether the above results demonstrated the functional synergism of SAM/Sugt1 in regulating SC proliferation. The conclusion was also substantiated when the expression dynamics and loss-of-function assays were performed using C2C12 MBs (Supplementary Fig. 5d–g). Interestingly, unlike SAM loss, Sugt1 knockdown did not seem to accelerate differentiation; instead, its loss may have delayed differentiation as assessed by the reduced number of MyoD+MyoG+ cells compared to control cells (Supplementary Fig. 5h).

To further ask how SAM association regulates Sugt1, we found that SAM depletion in SC did not alter the mRNA level of Sugt1 (Fig. 5j) or its proper localization at kinetochores in prometaphase (Supplementary Fig. 5i). Furthermore, it did not appear to alter the basal level of Sugt1 protein (Fig. 5k, left two lanes). However, treatment with a protein biosynthesis inhibitor, CHX, caused a marked decrease (31.4%) of Sugt1 protein in KO (Fig. 5k, lane 4 vs. 2) whereas only 18% in WT (Fig. 5k, lane 3 vs. 1), suggesting lower stability of Sugt1 in KO cells. Consistently, in a 20 h long CHX chase experiment, the half-life of Sugt1 protein was reduced at a faster rate upon SAM knockdown, confirming SAM is required for maintaining the protein stability of Sugt1 (Fig. 5l). Moreover, the decreased Sugt1 upon SAM depletion was blocked in the presence of a proteasome inhibitor, MG132 (Fig. 5m, lane 6 vs. 4), suggesting SAM may stabilize Sugt1 through preventing its ubiquitination. To further test this notion, HA-tagged ubiquitin protein was over expressed in Ctrl or shSAM MBs together with Sugt1 protein; we found an increased accumulation of poly-ubiquitinated Sugt1 in shSAM cells (Fig. 5n). Consistently, the stability of Sugt1 protein was rescued after restoring SAM expression in the presence of CHX without changing its RNA level (Supplementary Fig. 5j and k). To examine if SAM stabilizing Sugt1 protein specifically occurs in MB cells, we found no decrease in Sugt1 level in primary hepatocytes isolated from WT vs. KO mouse with or without CHX treatment (Supplementary Fig. 5l). Lastly, to further strengthen that SAM promotes MB proliferation through stabilizing Sugt1 protein, we found that overexpressing the WT or a stable mutant of Sugt1 (Sugt1-4A)[12] fully rescued the deficient proliferation of SAM KO cells while over-expressing a highly unstable mutant of Sugt1 (Sugt1-4E)[12] failed (Fig. 5o).

**SAM/Sugt1 regulate kinetochore assembly in MBs.** Since Sugt1 is critical for proper kinetochore assembly during cell division[10,41], we next tested whether SAM/Sugt1 together regulate SC proliferation through modulating kinetochore assembly. By staining chromosomes with DAPI, centromeres with ACA and

spindles with α-Tubulin, in control cells, a robust spindle structure was preserved in metaphase cells, and bundles of microtubules were observed to terminate in kinetochores (Fig. 6a). In contrast, cells with Sugt1 knockdown exhibited disorganized spindle structures with multipolar spindles and fragmented spindle poles frequently observed (Fig. 6a). The above phenomena were also observed in C2C12 MBs when Sugt1 was decreased (Supplementary Fig. 6a). Altogether, our data demonstrate Sugt1 is important for proper chromosomal alignment and spindle organization and thus timely mitotic division of MBs. Next, to demonstrate that SAM functions synergistically with Sugt1, we found SAM KO cells displayed evident defects in chromosome alignment and mitotic spindle formation (Fig. 6b) (Supplementary movies 1–4). Again, this was also more frequently observed in C2C12 MBs with SAM knockdown vs. control cells (Supplementary Fig. 6b), confirming SAM is needed for proper chromosomal alignment and mitotic division. To further determine if the above observed mitotic defects in SAM-depleted cells were due to kinetochore abnormalities, we examined kinetochore–microtubule (kt–mt) attachments under cold treatment considering the loss of cold stable kt–mt attachments is commonly used as an indicator of kinetochore defects[43]. WT and KO SCs were treated on ice for 10 min followed by α-Tubulin staining of microtubules (Fig. 6c); the fluorescence intensity was markedly decreased (20.76%) in KO vs. WT SCs, suggesting SAM loss led to increased instability of microtubules due to decreased kinetochores attaching. Consistently, when performed on C2C12, the same conclusion was reached; a reduced fluorescence intensity of microtubules was found in shSAM MBs under cold treatment (Supplementary Fig. 6c).

To further pinpoint the defect in kinetochore assembly upon SAM loss, we examined the localization of Mis12 complex, since it is known as a client of Hsp90-Sugt1 to be stabilized and targeted to the kinetochore[11]. As expected, by IF a lower level of fluorescent signals of Dsn1 subunit was observed at the kinetochores in the mitotic KO vs. WT cells (Fig. 6d). We next examined the localization of Hec1 giving that as a so-called keystone complex Mis12 contributes to the localization of Ndc80 complex[44]; interestingly, an over-accumulation of Hec1 kinetochore signals was detected in the KO vs. WT SCs (Fig. 6e). Similar phenomena were also observed in C2C12 MBs with decreased SAM knockdown (Supplementary Fig. 6d) despite the total level of Hec1 protein was largely unaltered (Supplementary Fig. 6e). Altogether, the above results confirmed the importance of SAM/Sugt1 in the proper localization of kinetochore components. Lastly, to pinpoint it is the defective kinetochore assembly that mediates SAM KO phenotype, we found that knockdown of Dsn1 or Hec1 in SCs also delayed cell proliferation as assessed by a decreased percentage of EdU+ cells (Figs. 6f, g and Supplementary Fig. 6f, g). Meanwhile, since Akt is a known client of Sugt1 and its phosphorylation at position 473 by Sugt1 can promote

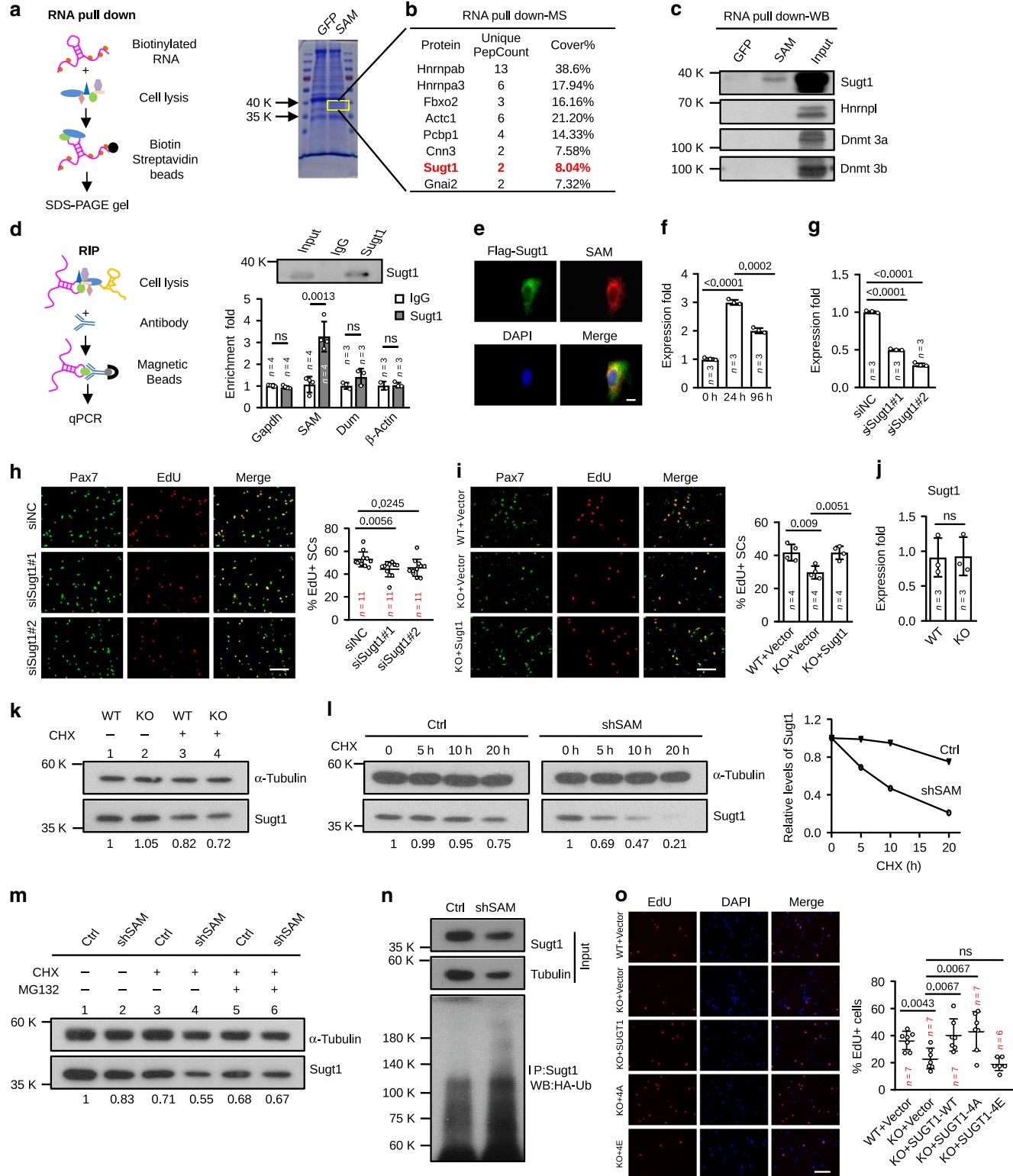

cancer cell proliferation[45], we tested if it could also mediate *SAM* effect but found that Akt p473 level was not decreased in KO vs. WT cells (Supplementary Fig. 6h). Altogether, the above findings demonstrate that *SAM* and Sugt1 together facilitate the assembly of kinetochore complex to ensure proper microtubule attachment in mitotic MBs. *SAM* deletion disrupts kinetochore assembly and thus delays the cell proliferation. Lastly, since it is believed that kinetochore disruption results in the mitotic arrest which is often

followed by cell death[46,47] or induces mitotic slippage accompanied by the production of aneuploid and cell senescence[48,49], we examined the consequence of such kinetochore defects in MBs and indeed detected an increased number of aneuploidy cells in KO vs. WT ASCs (Fig. 6h). However, no sign of cell apoptosis was detected earlier (Supplementary Fig. 4g); SA-β-Gal staining also revealed no indication of cellular senescence (Supplementary Fig. 6i).

**Fig. 5 _SAM_ binds and stabilizes Sugt1 in myoblasts. a** Left: Schematic of RNA pull-down assay. Right: The band (framed in the yellow box) was extracted for mass spectrometry (MS) analysis. **b** The partial list of proteins identified by MS. **c** Western blot (WB) confirmed the association of _SAM_ with Sugt1 but not Hnrnpl, Dnmt3a, and Dnmt3b proteins. 30 μg (or 1.5%) of cell lysate was used as input. **d** Left: Schematic of native RNA immunoprecipitation (RIP) assay. Right. WB analysis of Sugt1 protein after IP with Sugt1 antibody. qRT-PCR detection of the retrieved RNAs. **e** The co-localization of Flag-Sugt1 and _SAM_ by FISH coupled with IF. **f** Expression of _Sugt1_ in FISCs, ASCs, and DSCs. **g** _Sugt1_ was knocked down in ASCs by two different siRNA oligos. **h** EdU labeling in the above cells. The percentage of EdU+ cells was quantified. **i** EdU labeling in ASC transfected with a Vector or Sugt1 expressing plasmid. The percentage of EdU+ SCs was quantified. **j** Expression of _Sugt1_ in ASCs from WT vs. KO mice. **k** Sugt1 levels in WT and KO ASCs treated with cycloheximide (CHX) for 10 h. **l** Sugt1 in myoblasts transfected with Ctrl or sh_SAM_ oligos and treated with CHX for the indicated time. The degradation rates of Sugt1 are shown on the right. **m** Sugt1 levels in cells treated with CHX or/and MG132 for 12 h. **n** WB analysis of Sugt1 ubiquitination after IP with Sugt1 antibody in cells transfected with HA-Ub and Sugt1 expressing plasmids. **o** EdU labeling in WT and KO ASC transfected with a plasmid expressing SUGT1 wild type (WT), or stabilized SUGT1-4A, or unstable SUGT1-4E mutant. The percentage of EdU+ cells was quantified. The data are presented as mean ± SD in **d**, **f–j**, and **o**. The _p_ values by two-tailed unpaired _t_ test are indicated in **d**, **f–j**, and **o**, ns not significant. The total number of biologically independent samples are indicated in **d**, **f–j**, and **o**. Scale bars: 10 μm **e**, 100 μm **h**, **i**, and **o**. Source data are provided as a Source Data file.

## Discussion

In this study, we identified and characterized the functional role of a lncRNA, _SAM_, in regulating SC activity and muscle regeneration. Collectively, our findings suggest a model, in which _SAM_ regulates SC proliferation by binding with co-chaperon protein Sugt1 to facilitate the kinetochore assembly during mitosis, thereby governing the fidelity of cell division (Fig. 7). We infer that _SAM_ stabilizes Sugt1 protein through direct association; it thus facilitates the correct localization of Mis12 complex which is required for proper assembly of kinetochore and microtubule attachment during the mitotic progression of MB cells. Loss of _SAM_ in SCs leads to disrupted cell division and delayed proliferation, thus impairs muscle regeneration after acute or chronic muscle injuries.

Although initially identified in C2C12 muscle cells through integrating RNA-seq and ChIP-seq analyses, we expanded our study to SCs to show that _SAM_ is highly enriched in activated SCs. Moreover, gain or loss of function of _SAM_ in both C2C12 cells and ASCs altered cell proliferation. Extending the in vitro cell culture-based investigation, we provided extensive mouse genetic evidence to characterize _SAM_ function in vivo utilizing three KO mouse models: whole-body KO, SC-specific inducible KO (iKO) and mdx; _SAM_ dKO mice. Results from using all three models consistently supported a role for _SAM_ in regulating muscle regeneration after acute and chronic injuries. The KO-first strategy allowed us to delete _SAM_ without major disruption of the genomic region, therefore, avoiding the complication of disrupting a potential enhancer in this region. Analyzing the KO mice led to the observation that _SAM_ is not essential for mouse survival and fertility. However, the regeneration process of skeletal muscle after acute injury by BaCl$_2$ injection was evidently impaired in both KO and iKO mice. Nonetheless, in both models, the injured muscle eventually recovered completely from the injury, indicating that loss of _SAM_ delays but does not block the regeneration of skeletal muscle. In the third model, the dKO mice displayed much more severe dystrophic phenotypes characterized by extensive fibrosis compared to the mdx controls; this could be caused by the amplified proliferative defect due to repeated cycles of degeneration–regeneration that is typical of dystrophic muscles. Taken together, findings from using the three mouse models solidified the role of _SAM_ in regulating skeletal muscle regeneration in vivo, which adds genetic evidence for the functionality of lncRNAs in vivo.

Through identifying its interacting protein partners, we gained mechanistic insights into how _SAM_ regulates SC proliferation. Sugt1 was identified as a specific interacting partner with _SAM_ in MBs; their association is supported by results of RNA pull-down, native RIP, and co-localization assays. Furthermore, we showed that association with _SAM_ probably serves to stabilize Sugt1 as _SAM_ loss appeared to increase the ubiquitination level of Sugt1.

Consistently, a recent report[12] demonstrated that an E3 ligase, RNF41 regulates the ubiquitination of Sugt1 in a phosphorylation-dependent manner and PHLPP1 dephosphorylates Sugt1 to prevent it from associating with RNF41. In the future it may be worthy of the efforts to further investigate if _SAM_ may facilitate the homodimerization of Sugt1 or involve in the dephosphorylation of Sugt1 in MBs.

At the cellular level, we showed Sugt1 is required for kinetochore assembly as loss of Sugt1 in MBs led to typical defects associated with cell mitosis; for example, cells presented pronounced defects in kinetochore–microtubule attachment, spindle formation and chromosome misalignments, which is in line with what was observed in Hela cells. Thus, in both Hela and SCs, Sugt1 appears to exert a conserved function of regulating kinetochore assembly. Similarly, loss of _SAM_ largely photocopied the kinetochore abnormalities observed in Sugt1-depleted cells, leading us to conclude that _SAM_ and Sugt1 synergistically regulate Mis12 targeting and kinetochore assembly to control MB proliferation. Expectedly, Dsn1 kinetochore signals were significantly decreased in _SAM_ KO cells, in line with what was observed in Hela cells when Sugt1 was depleted. According to Davies et al. [11] the degradation of Dsn1 in Hela cells is dependent on Skp-Ub ligase thus suggesting this Ub pathway may be well functional in MB cells. Nevertheless, we observed increased accumulation of Hec1 protein at the kinetochores upon _SAM_ loss, indicating Hec1 may not be subject to Skp-Ub degradation in MBs. Still it was shown that over-accumulation of Hec1 in mouse MEF cells caused aberrant spindle phenotype[9], suggesting the mislocalization of Hec1 is indeed detrimental to the assembly of kinetochore and microtubule attachment in the _SAM_ KO cells.

It is also interesting to ponder on the fate of the MBs with the abnormality in cell division. In many studies, kinetochore defective cells will show arrest or delay in metaphase via the spindle assembly checkpoint (SAC), or perhaps slip out of mitosis with chromosome segregation errors increasing the frequency of senescence or apoptosis[47,49]. Indeed, an increased number of aneuploidy cells was detected upon _SAM_ loss, which did not lead to evident cell apoptosis or senescence; Unlike SCs deficient in SAC which resisted differentiation[50], _SAM_ loss did not seem to impede MB differentiation propensity. In fact, precocious differentiation was observed in _SAM_ KO cells, which seems to suggest that the aneuploidy MBs in _SAM_ KO may have eventually undergone premature differentiation. Coincidentally, Gogendeau et. al. [51] has described similar consequences in neural stem cells (NSCs) and intestine stem cells (ISCs), where they found that aneuploid NSCs do not die by apoptosis, instead, they display G1 lengthening and undergo premature differentiation[51]. Intriguingly, Sugt1 loss did not cause premature differentiation. We suspect it is possible that Sugt1 depletion had caused much more severe defects so that the cells eventually underwent apoptosis

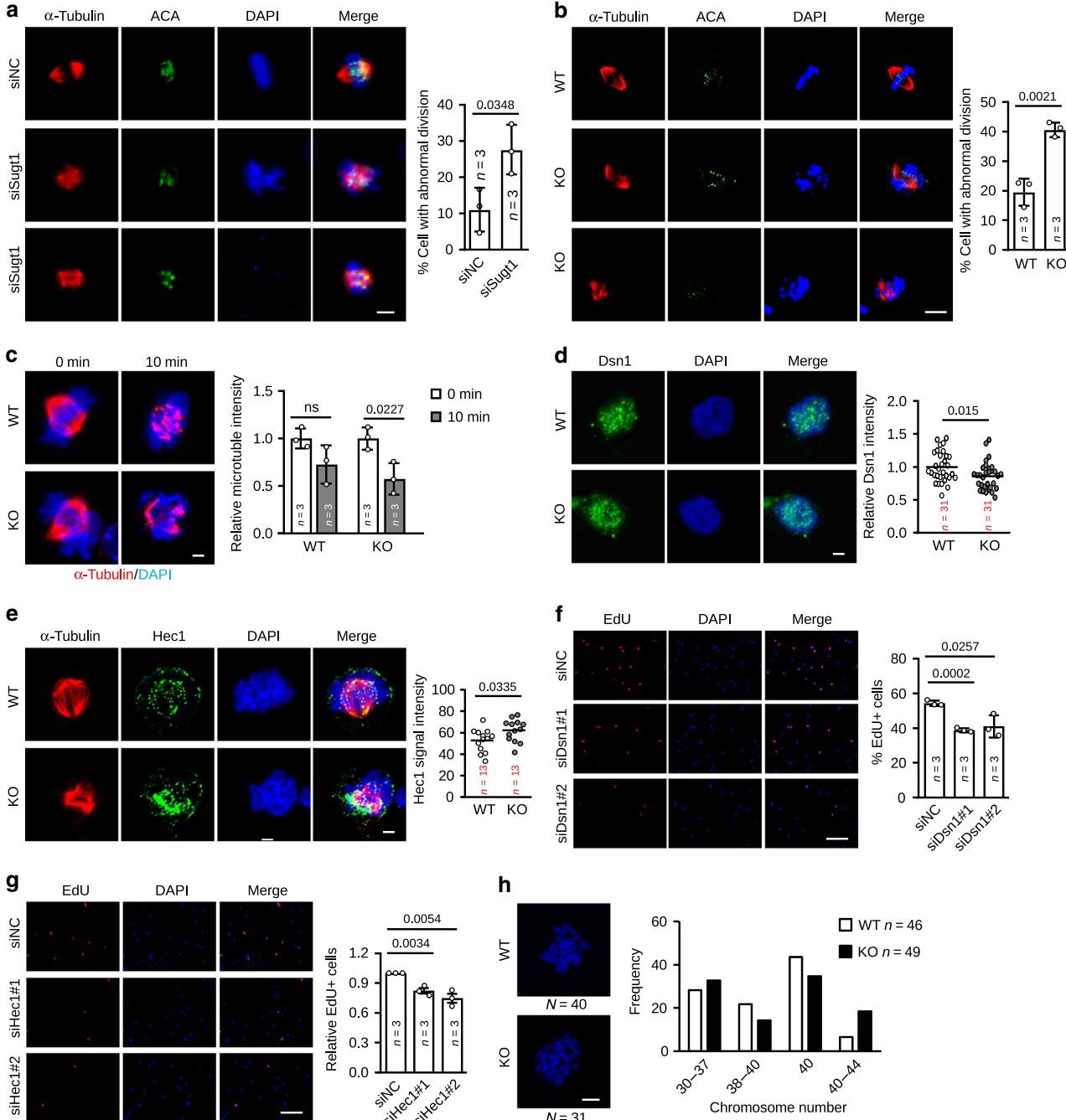

**Fig. 6 *SAM*/Sugt1 regulates kinetochore assembly and mitotic division in myoblasts. a** IF staining for α-Tubulin and ACA in ASC transfected with si*Sugt1* or negative control oligos. Cells in metaphase with mis localized ACA and non-bipolarized distribution of α-Tubulin were regarded as abnormally dividing cells and their percentage was quantified from at least 40 cells per group for each experiment. $n$ = the number of independent experiments. **b** The above assay was performed in WT and KO ASC and the quantification of abnormally dividing cells was conducted from at least 20 cells per group for each experiment. $n$ = the number of independent experiments. **c** The above SCs were cold treated on ice for the indicated time and stained for α-Tubulin. The average intensity of staining was measured from at least 6 cells per group for each experiment using in house script. $n$ = the number of independent experiments. **d** IF staining of Dsn1 or **e** Hec-1 and Tubulin were performed in cultured SCs from WT or KO mice. Cells were synchronized to the mitotic stage by nocodazole treatment for 3 h in **d**. Maximum Dsn1 and Hec1 fluorescent signals at kinetochores were quantified from the indicated number of cells. **f** EdU-labeling assay in ASCs transfected with si*Dsn1* or **g** si*Hec1*. The percentage of EdU+ SCs was quantified. **h** (Left) Representative images of DAPI-stained chromosome metaphase spreads from WT and KO SCs. (Right) Quantification of chromosome numbers of metaphase spreads from WT and KO SCs. The data are presented as mean ± SD in **a**–**c**, **f**, and **g**. The center line in **d** and **e** is presented as mean. The p values by two-tailed unpaired *t* test are indicated in **a**–**g**, ns not significant. The total number of independent experiments in **a**–**c**, **f**, **g** and biologically independent samples in **d**, **e**, and **h** are indicated. Scale bars: 5 μm **a** and **b**, 2 μm **c**–**e**, 100 μm **f** and **g**, 50 μm **h**. Source data are provided as a Source Data file.

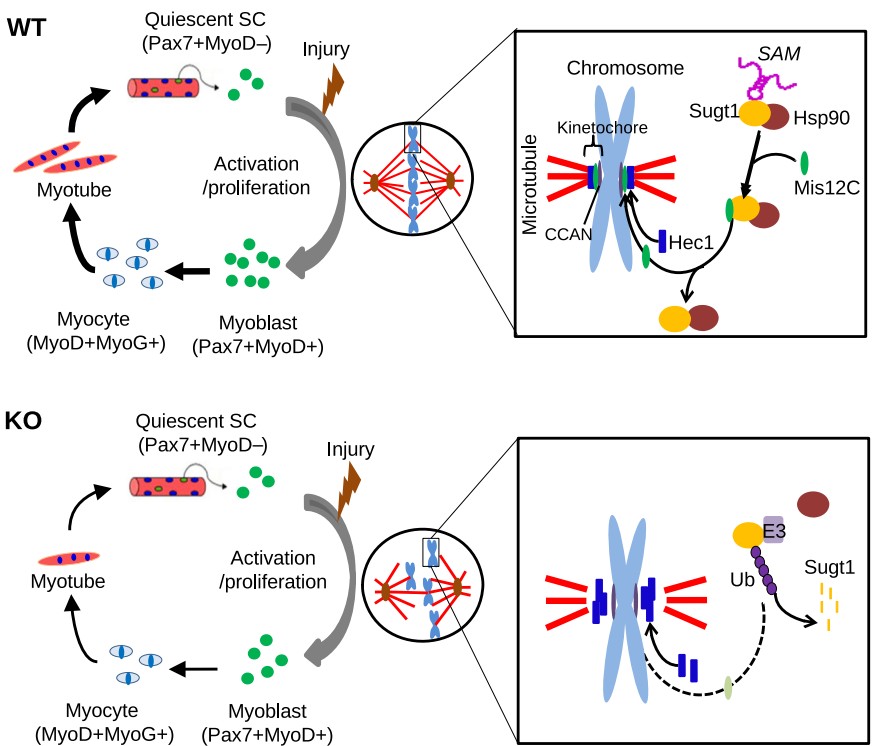

**Fig. 7 Schematic model depicting the functional role of *SAM* during SC activation/proliferation.** In WT mice, *SAM* regulates SCs proliferation by binding with co-chaperon protein Sugt1 to facilitate the kinetochore assembly during mitosis, thereby governing the fidelity of cell division. In KO mice, loss of *SAM* induces degradation of Sugt1 by ubiquitination then disrupts kinetochore assembly in mitotic cells due to mislocalization of Dsn1 and Hec1, delays proliferation and impairs muscle regeneration after acute or chronic muscle injuries.

without being able to differentiate; this needs to be further investigated using a Sugt1 KO mouse model. Alternatively, it is also likely that the differentiation function of *SAM* may not be fully dependent on Sugt1. Altogether, our findings add to the growing list of cellular mechanisms studied in the proliferation and differentiation of muscle stem cells and their progeny, thus enhancing our knowledge in the broad field of muscle stem cell and regeneration. Additionally, to the lncRNA field, this will add in vivo genetic evidence for lncRNA involvement in muscle regeneration and provides new insights into the mechanisms of lncRNA action to the growing list of lncRNA functions.

## Methods

**Mouse studies.** *SAM* KO heterozygote (*SAM*$^{-/+}$) mice (C57BL/6 background) were generated in the Model Animal Research Center of the Nanjing University (Nanjing, China). FLPeR mice were purchased in the Model Animal Research Center of the Nanjing University (Nanjing, China). *SAM* KO strain (*SAM* KO: *SAM*$^{-/-}$, littermate control: *SAM*$^{+/+}$) were housed in our laboratory animal services center at the Chinese University of Hong Kong (CUHK). Pax7$^{creER}$ mice were purchased from the Jackson Laboratory. Mdx mouse strains were purchased from the Jackson Laboratory. Pax7-nGFP mice[31] were gifts from by Prof. Shahragim Tajbakhsh (Institut Pasteur). *SAM* iKO strain (*SAM*-iKO: *SAM*$^{fl/fl}$; Pax7$^{creER/+}$, littermate control: *SAM*$^{+/+}$; Pax7$^{creER/+}$) were obtained by crossing *SAM* KO mice with FLPeR mice and Pax7$^{creER}$ mice. To induce Cre-mediated *SAM* deletion, TM (T5648, Sigma) was injected intraperitoneally at 2 mg per 20 g body weight for 5 days. *SAM*/mdx(dKO) strain (*SAM*-dKO: *SAM*$^{-/-}$; mdx, littermate control: *SAM*$^{+/+}$; mdx) were generated by crossing *SAM* KO mice with mdx mice. To induce acute muscle injury, 50 μl of 1.2% BaCl$_2$ (dissolved in sterile demineralized water) was injected into TA muscle of ~2 months old mice. Muscles were harvested at designated time points for further analysis. For EdU incorporation assay in vivo, one lower hindlimb muscle was subjected to 50 μl of 1.2% BaCl$_2$ injection. Then 10 mM EdU was injected intraperitoneally at 70 μl per 20 g body weight 2 days after injury, followed by FACS isolation of SCs 12 h later. Cells were then collected and fixed with 4% PFA. EdU-labeled cells were visualized using click chemistry with an Alexa Fluor® 594 conjugated azide. Pictures were captured with a fluorescence microscope (Leica). For all animal-based experiments, at least three pairs of littermates or age-matched mice were used. Primers for mice genotyping are listed in Supplementary Table 1. All animal experiments were performed in accordance with guidelines for experimentation with laboratory animals set in the Chinese University of Hong Kong (CUHK) and approved by the Animal Experimentation Ethics Committee of CUHK (Ref no. 15/027/MIS-6-U). The mice were maintained in animal room with 12 h light/12 h dark cycles, temperature (22–24 °C), humidity (40–60%) at animal facility in CUHK.

**Cell line culture and drug treatment.** Mouse C2C12 MB cells (CRL-1772) were obtained from American Type Culture Collection (ATCC) and cultured in growth medium, GM (DMEM medium (12800-017, Gibco) with 10% fetal bovine serum, FBS (10270-106, Gibco), 1% penicillin/streptomycin, P/S (15140-122, Gibco)), or differentiation medium, DM (DMEM medium with 2% horse serum (16050-114, Gibco), 1% P/S) in incubator at 37 °C. MG132 (M8699, Sigma, 10 μM) and CHX (Sigma, 100 μg ml$^{-1}$) were used for incubation for the indicated time.

**Satellite cell isolation and culture.** Hindlimb muscles from mice were digested with collagenase II (LS004177, Worthington, 1000 units ml$^{-1}$) for 90 min at 37 °C, the digested muscles were then washed in washing medium (Ham's F-10 medium (N6635, Sigma) containing 10% horse serum, heat-inactivated (HIHS, 26050088, Gibco), 1% P/S) before SCs were liberated by treating with Collagenase II (100 units ml$^{-1}$) and Dispase (17105-041, Gibco, 1.1 unit ml$^{-1}$) for 30 min. The suspensions were passed through a 20 G needle to release myofiber-associated SCs. Mononuclear cells were filtered with a 40-μm cell strainer and incubated with the following primary antibodies: Vcam1-biotin (105704, BioLegend), CD31-FITC (102506, BioLegend), CD45-FITC (103108, BioLegend), and Sca1-Alxa647 (108118, BioLegend). The Vcam1 signal was amplified with streptavidin-PE-cy7 (405206, BioLegend) or Streptavidin-PE (554061, BD Biosciences). All antibodies were used at a dilution of 1:75. The BD FACSAria Fusion Cell Sorter (BD Biosciences) was used for SC sorting following the manufacturer's instructions. BD FACSDiva (version 8.0.1, BD Biosciences) software is used to manage the setup, acquisition, and analysis of flow cytometry data. Coverslips and cultural wells were coated with poly-D-lysine solution (p0899, Sigma) at 37 °C for overnight and then coated with extracellular matrix (ECM) (E-1270, Sigma) at 4 °C for at least 6 h. FACS-isolated SCs were seeded in coated wells and cultured in Ham's F10 medium with 10% HIHS, 5 ng ml$^{-1}$ β-FGF (PHG0026, Thermo Fisher Scientific) and 1% P/S, or cultured in differentiation medium (DM) (Ham's F-10 medium containing 2% horse serum and 1% P/S).

**Single myofibers isolation and culture.** Briefly, EDL muscles were dissected and digested with Collagenase II (800 units ml$^{-1}$) in DMEM medium at 37 °C for 75 min. Single myofibers were released by gentle trituration with Ham's F-10 medium

containing 10% HIHS and 1% P/S) and cultured in this medium for designated time points.

**Cell proliferation, apoptosis, and cell-cycle analyses.** EdU incorporation assay was performed following the instruction of Click-iT® Plus EdU Alexa Fluor® 594 Imaging Kit (C10639, Thermo Fisher Scientific). Cells were incubated with 10 μM EdU for designated time before fixation. For MTS assay, cell growth rate was evaluated by using CellTiter 96® Aqueous One Solution Reagent Cell Proliferation Assay (MTS) kit (Promega, Madison, WI) according to the manufacturer's instruction. Generally, the cells were incubated with MTS for 3 h before absorbance measurement at 490 nm. Apoptosis was measured by TUNEL staining using the In-Situ Cell Death Detection Kit (Roche). For cell-cycle analysis, MBs were labeled with propidium iodide (PI) or Hoechst 33342 (5 μg ml$^{-1}$) for 45 min at 37 °C and sorted in the BD FACSVerse flow cytometer or BD FACSAria Fusion Cell Sorter. The results of cell cycle were analyzed using the WinMDI 2.8 software.

**SA-β-galactosidase staining.** Cellular senescence was evaluated by β-galactosidase activity using β-galactosidase Senescence Kit (#9860, Cell Signaling Technology). Briefly, cells were fixed for 15 min followed by washing in PBS twice. Then fixed cells were incubated with β-galactosidase staining solution at 37 °C in a dry incubator (no CO$_2$) at least overnight. The cells were then observed under a microscope for the development of blue color.

**Chromosome spread assay.** Cells were cultured for 3 days and treated with 100 ng ml$^{-1}$ nocodazole for 3 h before harvesting. Trypsinized cell pellets were resuspended in pre-warmed hypotonic solution (75 mM KCl) and incubated for 20 min at 37 °C followed by collecting by centrifugation for 5 min at 500 × g and gently resuspended with freshly prepared fixative solution (methanol/glacial acetic acid 3:1). Cells were fixed for 30 min. Two or three drops of suspended cells were released to pre-cold slides. The slides were then air-dried, and chromosomes were stained with DAPI.

**Isolation of mouse primary hepatocytes.** Liver tissue was isolated from mice and finely minced followed by digestion with collagenase II (400 U ml$^{-1}$) in water bath with shaking at 37 °C for 30 min. Digested tissue was mixed with a 10 ml serological pipette. The solution was triturated for 10–15 times or until the suspension traveled up and down the pipette smoothly without clogging. The cell suspension was then filtered through 70 μm cell strainer and centrifuged by 1300 rpm for 5 min. Cell pellet was washed twice in PBS and resuspended in culture medium (DMEM supplemented with 10% FBS, 100 U ml$^{-1}$ penicillin and 100 IU ml$^{-1}$ streptomycin). Primary hepatocytes were seeded on dishes and incubated at 37 °C with 5% CO$_2$ for 3 h. After cells had adhered (3–4 h) media was removed and replaced with fresh culture medium and continued to culture for 3 days.

**Plasmids.** Full-length mouse *Sugt1* was cloned into flag-tagged pcDNA3.1(+) vector (Life Technologies) between Kpn1 and Xbal1 sites. To construct *SAM* expression plasmid, full length of *SAM* was amplified and cloned into pcDNA3.1 (+) vector between Nhe1 and Kpn1 sites. Enhanced green fluorescent protein (GFP) was cloned into the XbaI site of pcDNA3.1(+) for in vitro transcription. *SAM* and *Sugt1* shRNAs were cloned into pSIREN Retro Q vector (Clontech). HA-Ub plasmid is a kind gift from Prof. Zhenguo Wu (Hong Kong University of Science and Technology, HKUST). SUGT-WT,4A,4E mutant plasmids are kind gifts from Prof. Subbareddy Maddika (Laboratory of Cell Death & Cell Survival, LCDCS, India)[12].

**Real-time PCR.** Total RNAs from tissues and cells were extracted using Trizol reagent (Invitrogen) following the manufacturer's instructions. cDNAs were prepared using HiScript® II Reverse Transcriptase Kit (Vazyme). SYBR™ Green master mixes (Life Technologies) and Light Cycler® 480 Real-Time PCR System (Roche) were used for quantitative real-time PCR (qRT-PCR) detection. *18s* and *Gapdh* were used for normalization. Primers for qRT-PCR are listed in Supplementary Table 1.

**Native RIP assay.** Native RIP assay was performed under physiological conditions without cross-linking[52]. Briefly, cell lysates were incubated overnight at 4 °C with antibody that were bound to Dynabeads protein G (Life Technologies) in NT2 buffer (50 mM Tris–HCl pH 7.4, 150,145 mM NaCl, 1 mM MgCl$_2$, and 0.05% NP40) containing 200 units RNaseOUT, 400 μM VRC, 10 μl of 100 mM DTT and 20 mM EDTA. Beads were then washed five times with NT2 buffer and treated with proteinase K for 30 min at 55 °C. RNAs were then isolated using the standard Trizol (Invitrogen) protocol and analyzed by qRT-PCR. Following antibodies were used in RIP assay: mouse anti-Sugt1 (sc-81822) and Normal mouse IgG (sc-2027).

**RNA pull-down assay.** Biotinylated RNAs were prepared using Biotin RNA Labeling Mix (Roche) and T7/T3 RNA in vitro transcription kit (Ambion). Fifteen micrograms of biotin-labeled RNAs were denatured at 90 °C for 2 min and then renatured with RNA structure buffer (10 mM Tris pH 7, 0.1 M KCl, 10 mM MgCl$_2$)

at RT for 20 min. Folded RNAs were mixed with 2 mg total protein lysate and incubated with 50 μl of Streptavidin agarose beads for one hour at room temperature (RT). After the incubation, beads were washed five times using RIPA buffer (50 mM Tris–HCl, pH 7.5, 150 mM NaCl, 1.0 mM EDTA, 0.1% SDS, 1% sodium deoxycholate, and 1% Triton X-100). Binding proteins were retrieved by boiling at 100 °C with loading buffer and further analyzed by running 10% SDS–PAGE gel according to the standard protocol. Proteins were detected by Coomassie Blue Staining using standard procedure and western blot.

**Mass spectrometry.** The band uniquely present in the *SAM* pull-done lane after Coomassie Blue staining was cut out and subject to LC–MS/MS analysis (Shanghai Applied Protein Technology, Shanghai, China).The MS scan was performed with the following parameters: positive ion detection; scan range ($m/z$) = 300–1800; resolution = 70,000 at 200$m/z$ automatic gain control (AGC) target = 1e6; maximum injection time = 50 ms; dynamic exclusion = 60 s. polypeptide and polypeptide fragments were collected according to the following parameters: after each full scan, 10 fragment maps (MS2 scan) were collected, MS2 Activation Type was HCD, isolation window was 2$m/z$, second-level mass spectral resolution was 17,500 at 200$m/z$, collision Energy was 30 eV, and underfill was 0.1%. The MS/MS spectra were searched with MASCOT engine (Matrix Science, version 2.2). The following option was used: peptide mass tolerance = 20 ppm, fragment mass tolerance = 0.1 Da, enzyme = trypsin, max missed cleavages = 2, fixed modification: carbamidomethyl (C), and variable modification: oxidation (M), acetyl (Protein N-term). The identified proteins were retrieved from the uniport mouse database (ref. no. 73952; download time: 20130313). Ion score ≥ 20. The number of unique peptides (Unique PepCount) and CoverPercent (Cover%: the number of detected amino acids/total number of amino acids in the protein) were used to identify proteins. In this study, one sample was analyzed once by LC–MS/MS.

**Western blotting.** Briefly, total proteins from cells were lysed in RIPA buffer supplemented with protease inhibitor cocktail, PIC (88266, Thermo Fisher Scientific) for 20 min on ice. The protein concentration was determined using a Bradford protein assay kit (Bio-Rad). The following antibodies and dilutions were used for western blot analysis. Mouse anti-Sugt1 (1:500, sc-81822, Santa Cruz), mouse anti-α-Tubulin (1:5000, B-5-1-2, Santa Cruz), mouse anti-Flag (1:1000, F1804, Sigma), mouse anti-Ub (1:5000, sc-8017, Santa Cruz), mouse anti-HA (1:1000, sc-7392, Santa Cruz), rabbit anti-Hnrnpl (1:1000, sc-28726, Santa Cruz), mouse anti-Dnmt 3a (1:1000, ab-13888, Abcam); rabbit anti-Dnmt 3b (1:1000, ab-2851, Abcam); and rabbit anti-Hec1 antibody[9] (1:5000) a very kind gift from Dr. Robert Benezra, Memorial Sloan Kettering Cancer Center, USA). The relative band intensities were quantified using ImageJ 1.50i (National Institutes of Health).

**Immunoprecipitation assays.** Cells were lysed with lysis buffer (50 mM Tris–HCl, pH 8.0, 150 mM NaCl, 0.1% SDS, 0.5% sodium deoxycholate, and 1% NP-40). The whole-cell lysates obtained by centrifugation (with equal concentration of protein in different samples) were incubated with 1 μg of Sugt1 antibody for overnight at 4 °C with rotation followed by binding to Dynabeads™ Protein G (Invitrogen) for 6 h at 4 °C. The immunocomplexes were then washed with washing buffer (10 mM Tris–HCl, pH 7.5, 150 mM NaCl, 1.0 mM EDTA, 1.0 mM EGTA, and 1% Triton X-100) four times and applied to SDS–PAGE.

**In vivo ubiquitination assay.** C2C12 cells were transfected with HA-ubiquitin and flag-Sugt1 plasmids. 38 h after transfection, cells were treated with MG132 (10 μM) for 10 h. The whole-cell extracts prepared by lysis buffer were subjected to immunoprecipitation of Sugt1 protein. The levels of ubiquitinated protein were then detected by immunoblotting with HA antibody.

**IF staining and image acquisition.** For IF staining, cells were fixed in 4% PFA for 15 min and permeabilized with 0.5% NP-40 for 10 min. Then cells were blocked in 5% BSA for 1 h followed by incubating with primary antibodies overnight at 4 °C and secondary antibodies for one hour at RT. For kinetochore protein staining, cells were need be pre-permeabilized in 1% Triton X-100 in PHEM buffer (60 mM Pipes, 25 mM HEPES, 10 mM EGTA, and 2 mM MgCl$_2$, pH 6.9) for 5 min before cells were fixed with 3.7% formaldehyde (Sigma) for 20 min. After fixation, cells were proceeded as described above. For cold-stable microtubule analysis, cells were incubated on ice for indicated times followed by fixation with PHEM buffer containing 3.7% formaldehyde and 0.2% Triton X-100 for 10 min on ice and cells then were stained as above. Antibodies and dilutions were used as following: rabbit anti-MyoD (1:100, Santa Cruz Biotechnology, Inc); rabbit anti-MyoG (1:200, Santa Cruz Biotechnology, Inc); mouse anti-Pax7 (1:100, Developmental Studies Hybridoma Bank); mouse anti-MF20 (1:50, Developmental Studies Hybridoma Bank); Donkey anti-Mouse IgG Alexa Fluor 488 or 594 (1:200, Invitrogen), Donkey anti-Rabbit IgG Alexa Fluor 594 (1:200, Invitrogen), goat anti-rabbit IgG Alexa Fluor 488 (1:200, Invitrogen); mouse anti-α-Tubulin (1:400, Santa Cruz), rabbit anti-Hec1 (1:200; a very kind gift from Robert Benezra, Memorial Sloan Kettering Cancer Center, USA), rabbit anti-Dsn1 (1:100; Biorbyt), and ACA (1:50, Antibodies Incorporated). All images were captured by a fluorescence microscope (Leica, DM 6000B) with Leica LAS AF software (LAS AF2.6.3) and laser scanning confocal microscope (Carl ZEISS LSM 880) with ZEN 2.3 (blue edition) software.

For measurements of fluorescence intensities, 10 optical slices were acquired at 0.3 μm intervals. Measurements of tubulin, Hec1, Dsn1, and Sugt1 intensities were conducted with maximum intensity projections of images by in house program written in MATLAB (R2014b) language. Exposure settings were held constant within each group of experiments.

**Immunohistochemistry**[53]. In brief, slides were fixed with 4% PFA for 15 min at RT and permeabilized in ice cold menthol for 6 min at −20 °C. Heat-mediated antigen retrieval with a 0.01 M citric acid (pH 6.0) was performed for 5 min in a microwave. After 4% BBBSA (4% IgG-free BSA in PBS; Jackson, 001-000-162) blocking, the sections were further blocked with unconjugated AffiniPure Fab Fragment (1:100 in PBS; Jackson, 115-007-003) for 30 min. The biotin-conjugated anti-mouse IgG (1:500 in 4% BBBSA, Jackson, 115-065-205) and Cy3-Streptavidin (1:1250 in 4% BBBSA, Jackson, 016-160-084) were used as secondary antibodies. Primary antibodies and dilutions were used as following: mouse anti-PAX7 (1:50, DSHB), mouse anti-MyoD (1:100, Dako, M3512), mouse anti-eMyHC (1:300, Leica, NCL-MHC-d), rabbit anti-Collagen1 (1:200; Novus, NBP1-30054), and rabbit anti-laminin (1:800, Sigma-Aldrich, L9393). Masson's trichrome staining was performed according to the manufacturer's instructions (ScyTek Laboratories, Logan, UT). All fluorescent images were captured with a fluorescence microscope (Leica, DM 6000B). Measurements of Collagen 1 and collagen positive area were conducted by in house ImageCount software written in MATLAB (R2014b) language.

**RNA fluorescence in situ hybridization**[54]. The Stellaris™-type oligonucleotides targeting SAM were modified with Biotin. Probe sequences are shown in Supplementary Table 2. Briefly, For SAM FISH, cells were fixed with 3.7% formaldehyde for 10 min at RT and permeabilized in 70% ethanol overnight at 4 °C and hybridized with probes in buffer (2× SSC, pH = 7.0, 10% formamide, 2 mM VRC, 0.2 mg ml$^{-1}$ BSA, 1 mg ml$^{-1}$ yeast tRNA, and 100 mg ml$^{-1}$ dextran sulfate) for overnight at 37 °C. After washing, cells were blocked with 4% BSA and then incubated with Cy3-streptavidin antibody (Jackson, ref: 016-160). Prolong Gold antifade reagent was applied to mount the slides for DAPI. Images were taken with a ×63 NA 1.4 oil objective on the laser scanning confocal microscope (Carl Zeiss LSM 880). For FISH and flag-Sugt1 IF co-staining, prior to the hybridization, cells fixed in 3.7% formaldehyde and stored in 70% ethanol were permeabilized with 0.5% Triton x-100 for 10 min at RT. After washing cells were proceeded with the FISH protocol as described above. The following antibodies and dilutions were used. Mouse anti-flag (1:200, Sigma). Goat anti–mouse IgG Alexa Fluor 488 (1:200, Invitrogen).

**RNA-seq and data analysis**. For library construction, we used a protocol as described before[13,14]. The purified library products were evaluated using a Bioanalyzer (Agilent) and SYBR qPCR and sequenced on an Illumina Hi-seq 2000 sequencer (pair-end with 50 bp). Sequenced fragments were mapped to reference mouse genome (mm9) using TopHat[54]. Cufflinks[55] was then used to estimate the relative abundance of transcripts in RNA-Seq experiments. Abundances were reported in fragments per kilobase per million (FPKM), which is conceptually analogous to the reads per kilobase per million (RPKM) used for single-end RNA-seq. Differentially expressed genes were identified if the fold change ≥ 1.5 by comparing siSAM and siNC samples.

**Statistics and reproducibility**. Data were analyzed using GraphPad Prism (version 8; GraphPad Software, San Diego, CA). Data were represented as the average of at least three biologically independent samples ± SD or ±SEM unless indicated. The statistical significance was assessed by the Student's two-tailed paired and unpaired t-test. ns, not significant. Representative images of at least three independent experiments were shown in Fig. 5a, c, d, e, k, m, n. and Supplementary Figs. 2b, e, f, l; 4g; 5d, e, 6e and h. Representative images of two independent experiments were shown in Supplementary Fig. 5b, k, and l.

**Reporting summary**. Further information on research design is available in the Nature Research Reporting Summary linked to this article.

## Data availability

The data supporting the findings of this study are available from the corresponding author on reasonable request. RNA-seq data have been deposited in the Gene Expression Omnibus under the accession code GSE126423. The mass spectrometry proteomics data have been deposited to the ProteomeXchange Consortium via the PRIDE[56] partner repository with the dataset identifier PXD018147. The source data underlying Figs. 1c–e, 1g–n, 2b, d, f, h–k, n, p, 3b, d, f, g, i, j, 4a–l, 5a, c, d, f–o, 6a–h and Supplementary Figs. 1b, d, f–t, 2b–d, g, h, j, l, 3a, b, 4b–f, 5b–l, 6a–i are provided in the Source Data file.

## Code availability

MATLAB language codes used in this study have been deposited on GitHub [https://github.com/jieyuanCUHK/SAM_paper].

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

## Acknowledgements

We thank Prof. Robert Benezra for his generous sharing the antibody of mouse Hec1; Prof. Subbareddy Maddika for sharing SUGT1-WT, SUGT1-4A, SUGT1-4E plasmids; Prof. Mara Brancaccio for sharing the Flag-Sugt1 plasmid; Prof. Ken Kaplan for his kind suggestions on kinetochore-related biology; Dr. Han Zhu and Prof. Tom. H. Cheung for their suggestions on fluorescence-activated cell sorting (FACS). This work was supported by General Research Funds (GRF) from the Research Grants Council (RGC) of the Hong Kong Special Administrative Region (14115319, 14133016, 14106117, and 14100018 to H.W.; 14116918 and 14120619 to H.S.); the National Natural Science Foundation of China (NSFC) to H.W. (Project code: 31871304), NSFC/RGC Joint Research Scheme to H.S. (Project code: N_CUHK 413/18); Focused Innovations Scheme: Scheme B to H.S. [Project Code: 1907307].

## Author contributions

Y.L. designed and performed most experiments, analyzed data, interpreted results, and drafted the manuscript. J.Y. performed image processing and analyzed RNA-seq data. S.Z. and F.C. performed individual mice experiments. Y.Z. and X.C. provided support and suggestions for FISH and RNA pulldown. L.L. and L.Z. provided individual cell experiments. C.Y.C. provided technique support during FACS isolation. H.S. and H.W. conceived the project, designed experiments, and wrote the manuscript.

## Competing interests

The authors declare no competing interests.
