## [Peer Review File · Nature Communications]

Reviewers' comments:

Reviewer #1 (Remarks to the Author):

This manuscript describes the identification and functional characterization of a non-coding RNA (SAM) that was highly expressed in proliferating myoblasts relative to quiescent satellite cells or differentiated myotubes. Using constitutive and conditional knockout mice and cell culture models, the authors show that SAM is necessary for proper assembly of mitotic apparatus (kinetochore and microtubule spindle poles) and therefore proliferation of myoblasts. Mechanistically, SAM was found to be associated with kinetochore protein Sugt1 and SAM KO led to increased ubiquitination of Sugt1. This is the first study demonstrating a role of non-coding RNA in kinetochore assembly in myoblasts, though the detailed molecular mechanism regarding how this RNA works is not completely clear. Overall the experiments are well performed, and the data largely support the conclusions. Some additional clarification may help strengthen the conclusions.

Most proteins associated with mitotic apparatus exhibit dynamic expression and localization during various stages of cell cycle. For SAM to function in kinetochore assembly, is SAM dynamically expressed at various cell cycle stages? At the population level, is SAM expressed in all or a subset of proliferating myoblasts?

It is intriguing that SAM KO does not affect embryonic myogenesis and generation of satellite cells given the extensive proliferation of myoblasts during development. The potential different functions of SAM in embryonic and postnatal myoblasts is interesting and should be explored.

The mild regenerative phenotype (described as a delay in regeneration) of the KO and cKO suggest that SAM is not absolutely required for the kinetochore organization and proliferation of myoblasts. This is in contrast to the massive defects in kinetochore and spindle pore organization observed in the cell culture studies (By the way the authors should provide some 3-D images to better illustrate spindle pore and kinetochore defects). Is this defect also observed in vivo? Abnormal kinetochore assembly usually leads to mitotic arrest and apoptosis. In this regard, the lack of apoptosis in the SAM KO cells (~40% exhibited kinetochore assembly defects) is puzzling. Is the cell cycle synchronized for the results in Figure 6? If not, is it possible that the various spindle pole structures represent different cell cycle stages?

Did the conditional KO (in which exons 3-4 are missing in satellite cells) generate a truncated transcript? Given the similar phenotypes in the constitutive (which prevented SAM transcription in all cells) and conditional KOs, it seems that exon 3-4 are essential for the function of SAM. Some domain analysis (or structural prediction) may provide insight into how SAM and Sugt1 interact.

As SAM KO increases polyubiquitination of Sugt1, it is intriguing why the protein levels of Sugt1 are not decreased in the SAM KO myoblasts. Also, is it possible to perform immunostaining to examine how SAM KO affect subcellular localization of Sugt1?

Minor:

As there are more than 20 publications on SAM homolog in humans, the human Lnc-RNA symbol (SNHG15) should be adopted to avoid confusion in nomenclature.

Page 3: please provide full names of various proteins in the kinetochore complex.

Page 8: “three RT-PCR primers targeting different regions (exon 1-2, exon 2-3 and exon 3-4) were used to detect possible transcription (Fig. 2a); no transcripts were detected with any pair of primers (Supplementary Fig. 2c) or in the isolated SCs (Fig. 2b), ...” As the polyA insertion was after exon 2, why did primers targeting exon 1-2 detect any transcripts? Is it due to active non-sense mediated RNA decay?

Page 13: Although the RNAseq results support the observed phenotype, they do not necessarily support the mechanism put forward later in the study. If SAM stabilizes Sugt1 and facilitate kinetochore assembly, then it should not affect expression of genes involved in cell cycle (for example SAM KO did not affect mRNA level of Sugt1). Even if it does affect gene expression, shouldn't SAM KO lead to an upregulation of cell cycle genes as a feedback mechanism?

Reviewer #2 (Remarks to the Author):

In the manuscript entitled, “Long non-coding RNA SAM promotes myoblast proliferation and skeletal muscle regeneration through Sugt1 and facilitating kinetochore assembly”, the authors characterize a novel RNA that interacts with the co-chaperone, Sugt1 (AKA Sgt1) and promotes muscle stem cell proliferation in favor of differentiation, a balance necessary for proper muscle regeneration. This work describes two major novel findings that are of interest to a wide cross-section of researchers who study cell division/chromosome segregation, protein complex assembly, cell differentiation as well as muscle regeneration: (i) first is the novel observation that a lncRNA can bind to and regulate a protein-chaperone complex and (ii) second is that this pathway plays a very interesting role in balancing muscle stem cell proliferation with myoblast differentiation. The connection between an lncRNA and a chaperone network is indeed novel and very interesting. I am particularly enthusiastic about this aspect of the work, although I have reservations about the authors ability to conclude that it is the kinetochore defects per se that result in the bias to away from cell proliferation to cell differentiation. My concern arises from the likelihood that the Hsp90-Sugt1 chaperone clients are diverse and thus the kinetochore defect, though real, represents a false correlation with the myoblast behavior. The authors themselves note that loss of SAM results in a wide range of transcriptional changes that are unlikely to result simply from mis-assembled kinetochores. Furthermore, the authors suggest in the Discussion that there are other SAM dependent factors (i.e., PRC2) that may affect differentiation at the transcriptional level, and the literature identifies AKT phosphorylation as downstream of Sugt1. I have made suggestions below to help address this complexity, which as it stands, detracts from the novelty of the result. I recommend

publication if these issues can be satisfactorily addressed.

Major issues to be addressed:

As mentioned above, the exact cause-effect relationship between kinetochore assembly and cell differentiation is not sufficiently addressed, and it is likely that the pleiotropic role of Sugt1/Hsp90 may affect more than just kinetochore assembly in myoblasts. One test of this hypothesis would be to directly target kinetochore assembly using an siRNA approach against Hec1/Ndc80 (or against Mis12, see #2 below). Based on the authors' model, directly disrupting kinetochore function should also delay the cell cycle and induce differentiation at a higher frequency in the satellite muscle cell cultures. This is an absolutely required experiment, especially given that the authors claim in the title of the paper that SAM promotes skeletal muscle regeneration through facilitating kinetochore assembly.

Another aspect of the authors' analysis that raises concerns about the kinetochore-assembly model is their observation that cells with reduced SAM levels enrich in G1. One might reasonably expect kinetochore defective cells to arrest or delay in metaphase via the spindle assembly checkpoint, or perhaps slip out of mitosis with chromosome segregation errors increasing the frequency of senescence or apoptosis. The wording in the text suggests there's no increase in apoptosis, though I cannot tell if this was assayed in cells that were expected to be undergoing cell division. Is there evidence of aneuploidy (e.g, polyploidy/tetraploidy from failed mitoses; stain with centromeric markers and count pairs) or senescence in SAM deficient myoblasts?

The current model of Sugt1/Sgt1 is that it acts as a co-chaperone for Hsp90, specifically linking Hsp90 to a subset of "clients". The data put forth by the authors suggests that SAM stabilizes Sugt1/Sgt1 and thus one might arguably stabilize the Hsp90-Sugt1/Sgt1 client or, as published previously, the Mis12 complex of the kinetochore. Given that there is specific client data, the authors should also measure the integrity of Mis12 complex under wild type and SAM KO conditions. The prediction is that it's the disruption of the Mis12 complex that ultimately impacts the quality of the Hec1/Ndc80 complex. This experiment is important as it connects the authors' model with known clients of the Hsp90-Sugt1/Sgt1 co-chaperone complex.

The other client based on the literature and highly related to the phenotype observed here, is Akt. Gao et al, (Mol Biol Rep) report that loss of Sgt1 in cancer cells inhibits cell proliferation through dephosphorylation of Akt at position 473 (growth promoting). Deletion of SAM and destabilization of Sugt1 in myoblasts might be predicted to decrease Akt P-473 and thus inhibit myoblast proliferation. A Western blot using the phospho-specific antibody to Akt P473 would allow the authors to narrow the client range of Sugt1 in myoblasts or implicate a much more relevant pathway that would better explain the G1 enrichment in SAM knockouts.

The lack of developmental defects in animals with a SAM KO suggests that its role in stabilizing Sugt1 might be specific to myoblasts. This raises the possibility that Sugt1 is uniquely unstable in myoblasts compared to other tissues or in muscle cells during development. Minimally, it would be clarifying to test the Ub-modification or stability of Sugt1 in cells NOT involved in muscle regeneration in the SAM KO animals. This seems like a missing control to their Sugt1 levels experiment in Figure 5.

Other comments:

Sugt1/Sgt1 dependent assembly of kinetochore complexes was described in HeLa cells (Davies et al, JCB

2010). Inhibition of Sugt1/Sgt1 resulted in degradation of mis-assembled kinetochore complexes, a turnover that was dependent on Skp1-Ub ligases. After siRNA of both Sugt1/Sgt1 and Skp1, non-functional kinetochore complexes accumulated at centromeres, much as the authors see in myoblasts deficient in SAM. The authors' findings suggest that Ub-mediated degradation of kinetochore complexes might also be down-regulated in the SAM deficient cells, as mis-assembled kinetochore complexes appear to be associated with the centromeres. The authors should make this relationship clearer when pointing out that the decrease in Sugt1/Sgt1 doesn't match the published results. One might predict that the Skp1-Ub ligase complex is also destabilized by loss of SAM, or is inherently low in myoblasts. This is first referred to on page 16 where the authors note the over-accumulation of Hec1/Ndc80, and then again in the Discussion on page 19.

On page 16, the authors describe cold treatment of cells to measure Kt-Mt stability. Their sentence makes it seem as though SAM depleted cells had already been suggested to lack of stable kinetochore-microtubule attachments by Rieder, C. Et al. This sentence needs to be re-written to make it clear that loss of cold stable kt-mt attachments is an indication of kinetochore defects.

Page 3, second paragraph: "Cell cycle involves" should read "The cell cycle involves". On the same line, "A faithful chromosome segregation". On the next line, "In vertebrates, kinetochore" should read, "In vertebrates, the kinetochore".

Page 4, the authors spend some time discussing lncRNAs found associated with centromere function, but I'm not clear why this is relevant to their work. It makes the reader anticipate that SAM will be associated with centromeric heterochromatin, but this is not the case. I suggest eliminating or reducing this paragraph.

Page 7, second paragraph, the "pro-proliferating function of SAM" is awkward and inaccurate. The authors mean to say that SAM is required for efficient myoblast proliferation".

Page 11, first paragraph, "However, when examining more closely" should read, "However, when examined more closely".

Page 12, second paragraph, the comparison of MyoG+ and MyoD+ sentence is missing the KO vs. wild type text.

Overall the materials and methods, as well as the statistical analysis performed, are acceptable, and would allow a research with the proper expertise in muscle SC culturing to reproduce the experiments.

Reviewer #3 (Remarks to the Author):

This study reports the isolation and characterization of a lncRNA (SAM) expressed in activated satellite cells and proliferating myoblasts. Both cell culture experiments and generation of two independent SAM knock-out mice indicate a role for SAM in promoting cell proliferation and inhibiting myoblasts fusion and differentiation. Biochemical experiments permitted the identification of putative SAM-interacting proteins. Sug1 (suppressor of G2 allele of SKP1) was, among others, purified from C2C12 myoblasts lysates using biotin-labeled SAM transcripts. SAM-knock out cells displayed reduced Sug1 protein half-life, suggesting a role of SAM in regulating Sug1 protein stability. Sug1 knock-down experiments revealed defects in spindle formation and mitotic division. Similar defects were observed in SAM-knock

out cells.

The phenotype observed in SAM-depleted cells and SAM-knock out mice are well documented. How SAM mechanistically regulates myogenesis remains to be firmly established. The SAM phenotypes are transient (3-4 days following injury) and not observed 1 month after injury.

1. Differences in WT and SAM KO cells are marginal, though statistically significant.

Figure 4d reports a 6% (91.78% vs 85.61%) difference in Pax7+ MyoD+ cells in WT and SAM KO mice.

Figure 4F indicates a 5.59% variation in Pax7+ MyoD + in WT and iKO cells.

2. The RNA pull-down reported in Figure 5C and Supplemental Figure 7 document an apparent discrepancy in the Input lanes for Sug1 (obviously saturated) and those for the remaining proteins. It is important for the authors to indicate the quantity of lysate (milligrams) employed in each experiment.

3. It is not clear how SAM prevents myoblast differentiation. Do Sug1 knock-down cells display anticipated differentiation as SAM-knock out cells?

4. Figure 2. eMyHC+ fibers decreased by 21.95% 4 days after injury. Was this difference observed at later times (14 days, 28 days following injury)?

Reviewer #1:

This manuscript describes the identification and functional characterization of a non-coding RNA (SAM) that was highly expressed in proliferating myoblasts relative to quiescent satellite cells or differentiated myotubes. Using constitutive and conditional knockout mice and cell culture models, the authors show that SAM is necessary for proper assembly of mitotic apparatus (kinetochore and microtubule spindle poles) and therefore proliferation of myoblasts. Mechanistically, SAM was found to be associated with kinetochore protein Sugt1 and SAM KO led to increased ubiquitination of Sugt1. This is the first study demonstrating a role of non-coding RNA in kinetochore assembly in myoblasts, though the detailed molecular mechanism regarding how this RNA works is not completely clear. Overall the experiments are well performed, and the data largely support the conclusions. Some additional clarification may help strengthen the conclusions.

1.1. Most proteins associated with mitotic apparatus exhibit dynamic expression and localization during various stages of cell cycle. For SAM to function in kinetochore assembly, is SAM dynamically expressed at various cell cycle stages? At the population level, is SAM expressed in all or a subset of proliferating myoblasts?

We thank the reviewer for the thoughtful comments. We agree results from the suggested experiments will strengthen our main conclusions. To answer whether SAM is dynamically expressed at various cell

cycle stages, we have now collected myoblast cells in G1/G0, S and G2/M phases of the cell cycle. Analysis of the RNA content at each phase revealed no significant difference in *SAM* expression throughout the cell cycle. This is consistent with what was shown by Liu XS et. al¹ that Sugt1 protein is not dynamically expressed at various cell cycle stages, instead it is mainly phosphorylated and transiently localizes at the kinetochores during prometaphase. The data can be found in Supplementary Fig. 1n and Page 9 of the revised text. To answer if *SAM* is expressed in all or a subset of proliferating myoblasts, we examined its expression in Pax7^{Hi} vs Pax7^{Lo} subpopulations² and found *SAM* expression did not significantly differ in these two subpopulations of myoblasts. The data can be found in Supplementary Fig. 1c and Pages 7 and 8 in the revised text.

1.2. It is intriguing that SAM KO does not affect embryonic myogenesis and generation of satellite cells given the extensive proliferation of myoblasts during development. The potential different functions of SAM in embryonic and postnatal myoblasts is interesting and should be explored.

We agree with the reviewer that the study can be extended to include embryonic and postnatal myogenesis even though the current study is focused on adult muscle regeneration. To get an idea of the possible *SAM* involvement in embryonic or postnatal myoblasts, we have now examined on embryonic day 18.5 or postnatal day 7 and found there was no overt changes in muscle morphology and the number of Pax7 positive cells in WT vs and KO mice, suggesting that *SAM* loss may not have impact on embryonic or postnatal myogenesis. Nevertheless, we acknowledge that in the future a more extensive study is needed to elucidate the different functionality of *SAM* in various stages of myogenesis. The new data can be found in Supplementary Fig. 2f-2g and 2h as well as Page 10 of the revised text.

1.3. The mild regenerative phenotype (described as a delay in regeneration) of the KO and cKO suggest that SAM is not absolutely required for the kinetochore organization and proliferation of myoblasts. This is in contrast to the massive defects in kinetochore and spindle pore organization observed in the cell culture studies (By the way the authors should provide some 3-D images to better illustrate spindle pore and kinetochore defects). Is this defect also observed in vivo? Abnormal kinetochore assembly usually leads to mitotic arrest and apoptosis. In this regard, the lack of apoptosis in the SAM KO cells (~40% exhibited kinetochore assembly defects) is puzzling. Is the cell cycle synchronized for the results in Figure 6? If not, is it possible that the various spindle pole structures represent different cell cycle stages?

We thank the reviewer for the critical comments. According to the suggestion, (1) we have now provided 3D images to better illustrate the spindle pole defects in supplementary videos. (2) We agree it will strengthen our conclusion if we can observe the kinetochore defects in vivo. To this end, we have attempted to observe kinetochore and spindle organization in SCs associated with freshly isolated myofibers. Unfortunately, we found it was impossible to capture SCs at metaphase on this ex vivo model. (3) The cells in Fig. 6a-c and 6e were not synchronized, as the synchronizing reagent (such as nocodazole and colchicine) will disrupt microtubules, making it impossible to examine the spindle poles.

We have gone back to re-examine the spindle pole structures carefully and believe the misshaped poles observed in the mutant are not variants associated with cell cycle stages because these diffusive, multi or monopoles are typically associated with abnormal cell division. In addition, we have now performed additional assays to demonstrate the abnormal kinetochore assembly in KO cells; for example, in Fig. 6c, we showed that cold stable kinetochore-microtubule attachments were decreased in KO vs Ctrl cells; also, in Fig 6e we showed abnormal accumulation of kinetochore protein Hec1 and in Fig. 6d

decreased signal of kinetochore protein Dsn1 in KO cells. Moreover, requested by Reviewer 2's comment 2.1, we have now showed that knockdown of kinetochore component Hec1 and Dsn1 delayed cell cycle (Fig. 6f-6g, Supplementary Fig. 6f and 6g), pinpointing kinetochore defects as the cause of cell proliferation defects. Altogether, we believe that loss of *SAM* leads to abnormal cell division by affecting kinetochore assembly.

We agree that abnormal kinetochore assembly usually leads to mitotic arrest and apoptosis, however, we now believe in our case it may have led to premature differentiation; in fact, Gogendeau, D et. al. has described similar consequences in neural stem cells (NSCs) and intestine stem cells (ISCs). They found that aneuploidy NSCs do not die by apoptosis³, instead, aneuploidy NSCs display G1 lengthening and undergo premature differentiation. We have now confirmed the presence of aneuploidy SCs in KO (Fig. 6h) and concluded these cells underwent premature differentiation, which may explain the relatively mild regenerative phenotype in one-injury induced regeneration. However, the phenotype was much more pronounced in dKO that underwent chronic regeneration allowing the defects in SCs to be amplified. We have revised the text on Pages 18, 19, 22 and 23 to include the above discussed points. Please also see our answer to Reviewer 2's comment 2.2.

1.4. Did the conditional KO (in which exons 3-4 are missing in satellite cells) generate a truncated transcript? Given the similar phenotypes in the constitutive (which prevented SAM transcription in all cells) and conditional KOs, it seems that exon 3-4 are essential for the function of SAM. Some domainial analysis (or structural prediction) may provide insight into how SAM and Sugt1 interact.

We thank the reviewer for the great suggestion. To detect any possible truncated transcripts, we have now designed RT-PCR primers targeting different regions (exon 1-2, exon 2-3 and exon 3-4) (Fig. 2a) and found a truncated transcript was in fact generated from exon1 to 2 (Fig. 2n) in the isolated SCs from iKO mice, suggesting the truncated transcript exon 1-2 may not be functional while exon 3-4 are essential for the function of SAM. According to the suggestion, we have also performed domain analysis and interestingly both 5' and 3' ends of SAM can retrieve Sugt1 with comparable efficiency as the full-length transcript. The newly added results can be found in Fig. 2n, Supplementary Figs. 5a and b as well as Pages 11 and 16 of the revised text

1.5. As SAM KO increases polyubiquitination of Sugt1, it is intriguing why the protein levels of Sugt1 are not decreased in the SAM KO myoblasts. Also, is it possible to perform immunostaining to examine how SAM KO affect subcellular localization of Sugt1?

Thanks for the comment and suggestion. Indeed, Sugt1 protein didn't seem to decrease in *SAM* KO vs Ctrl myoblasts when cells were cultured for 2 days. We have now examined SMA protein level in the myoblasts that were cultured for 4 days when the Sugt1 expression reached a higher level; and we found a slight decrease of Sugt1 in KO vs Ctrl cells (Supplementary Fig. 6h). However, the decrease of Sugt1 under treatment of CHX which blocks protein synthesis is very evident in repeated assays. We thus speculate that the synthesis of Sugt1 is somehow accelerated in KO vs Ctrl cells to compensate for the increased degradation rate. Meanwhile, according to the suggestion, we also have performed the immunofluorescence staining of Sugt1 and found that *SAM* loss did not affect subcellular localization of Sugt1 in mitotic stage. We believe that *SAM* binding with Sugt1 is to stabilize the protein and facilitate the targeting of Mis12 complex to the kinetochore. The newly added result can be found in Supplementary Fig. 5i as well as Page 16 of the revised text.

Minor:

1.6. As there are more than 20 publications on SAM homolog in humans, the human Lnc-RNA symbol (SNHG15) should be adopted to avoid confusion in nomenclature.

Thank you for your comment. We think the name *SAM*, Sugt1 Associated Muscle lincRNA can better reflect its function uncovered in our study. There are examples of lincRNAs with different names in human and mouse. For example, human lincRNA MEG3 is also called Gtl2 in mouse. We thus propose to keep *SAM* while clarifying in the manuscript it is the homolog for human SNHG15 on Page 2. But we are flexible on this if the reviewer and editor deem it is better to use the human name.

1.7. Page 3: please provide full names of various proteins in the kinetochore complex.

Thank you for your comment. We have now added full names of various proteins on Page 3.

1.8. Page 8: “three RT-PCR primers targeting different regions (exon 1-2, exon 2-3 and exon 3-4) were used to detect possible transcription (Fig. 2a); no transcripts were detected with any pair of primers (Supplementary Fig. 2c) or in the isolated SCs (Fig. 2b), ...” As the polyA insertion was after exon 2, why did primers targeting exon 1-2 detect any transcripts? Is it due to active non-sense mediated RNA decay?

Thanks for the comment. To confirm whether the loss of transcripts of exon1-2 in KO mice is due to activation of the non-sense mediated RNA decay, we have now treated the cells with cycloheximide (CHX) which has a general capacity to reverse non-sense mediated RNA decay⁴. Indeed, we found the treatment has led to increased transcript of exon1-2 in KO SC, indicating active non-sense mediated RNA decay could be responsible for the loss of the transcript. The data can be found in Supplementary Fig. 2d and on Page 10 of the revised text.

1.9. Page 13: Although the RNAseq results support the observed phenotype, they do not necessarily support the mechanism put forward later in the study. If SAM stabilizes Sugt1 and facilitate kinetochore assembly, then it should not affect expression of genes involved in cell cycle (for example SAM KO did not affect mRNA level of Sugt1). Even if it does affect gene expression, shouldn't SAM KO lead to an upregulation of cell cycle genes as a feedback mechanism?

Thanks for the thoughtful comment. We agree RNA-seq cannot be used to provide any mechanistic insight; in our study it is in fact used to confirm the cell cycle defect caused by *SAM* loss. We have revised the text on Page 14 to avoid the misunderstanding. In fact, prior reports have shown that cell cycle genes can be altered in cells with kinetochore defects; for example, Wootae Kim et al.⁵ has shown that the cell cycle genes can be down-regulated upon kinetochore defects, similar to what we observed.

Reviewer #2:

In the manuscript entitled, “Long non-coding RNA SAM promotes myoblast proliferation and skeletal muscle regeneration through Sugt1 and facilitating kinetochore assembly”, the authors characterize a novel RNA that interacts with the co-chaperone, Sugt1 (AKA Sgt1) and promotes muscle stem cell

proliferation in favor of differentiation, a balance necessary for proper muscle regeneration. This work describes two major novel findings that are of interest to a wide cross-section of researchers who study cell division/chromosome segregation, protein complex assembly, cell differentiation as well as muscle regeneration: (i) first is the novel observation that a lncRNA can bind to and regulate a protein-chaperone complex and (ii) second is that this pathway plays a very interesting role in balancing muscle stem cell proliferation with myoblast differentiation. The connection between an lncRNA and a chaperone network is indeed novel and very interesting. I am particularly enthusiastic about this aspect of the work, although I have reservations about the authors ability to conclude that it is the kinetochore defects per se that result in the bias to away from cell proliferation to cell differentiation. My concern arises from the likelihood that the Hsp90-Sugt1 chaperone clients are diverse and thus the kinetochore defect, though real, represents a false correlation with the myoblast behavior. The authors themselves note that loss of SAM results in a wide range of transcriptional changes that are unlikely to result simply from mis-assembled kinetochores. Furthermore, the authors suggest in the Discussion that there are other SAM dependent factors (i.e., PRC2) that may affect differentiation at the transcriptional level, and the literature identifies AKT phosphorylation as downstream of Sugt1. I have made suggestions below to help address this complexity, which as it stands, detracts from the novelty of the result. I recommend publication if these issues can be satisfactorily addressed.

Major issues to be addressed:

2.1. As mentioned above, the exact cause-effect relationship between kinetochore assembly and cell differentiation is not sufficiently addressed, and it is likely that the pleiotropic role of Sugt1/Hsp90 may affect more than just kinetochore assembly in myoblasts. One test of this hypothesis would be to directly target kinetochore assembly using an siRNA approach against Hec1/Ndc80 (or against Mis12, see #2 below). Based on the authors' model, directly disrupting kinetochore function should also delay the cell cycle and induce differentiation at a higher frequency in the satellite muscle cell cultures. This is an absolutely required experiment, especially given that the authors claim in the title of the paper that SAM promotes skeletal muscle regeneration through facilitating kinetochore assembly.

We thank for the positive comment on the novelty of our study and we fully agree that the exact cause-effect relationship between kinetochore assembly and cell differentiation needs to be sufficiently addressed. Since the pleiotropic role of Sugt1/Hsp90 may affect more than just kinetochore assembly thus it is necessary to confirm it is the kinetochore defect that mediates SAM effects on myoblast proliferation. To this end, following the suggestion, we have now performed siRNA knock down assay for Hec-1, Dsn1 or Sugt1 and found this indeed delayed cell proliferation (Fig. 6f, 6g and 5h), thus solidifying the conclusion that SAM/Sugt1 regulation of myoblast proliferation is mediated through the kinetochore assembly function of Sugt1. However, we wish to point out that the premature differentiation phenotype observed in SAM KO was not seen upon Sugt1 depletion (Supplementary Fig. 5h). We suspect that this is because Sugt1 depletion had caused more severe kinetochore assembly defects that led to complete blockage of cell proliferation and eventually cell apoptosis based on our preliminary observations in culture. It remains to be fully investigated using the Sugt1 knockout mouse that are being generated in our lab. Alternatively, it is also likely that SAM may have Sugt1 independent role in inhibiting myoblast differentiation, which is also supported by limited preliminary data but needs to be further teased out. We thus decide that we should focus on the proliferative impact of SAM/Sugt1 in myoblasts in the current manuscript. Please also see the answer below to comment 2.2 and the answer

to Reviewer 1's comment 1.3. The newly added results can be found in Figs. 6f, 6g, Supplementary Fig. 5h, 6f and 6g and Pages 16 and 19 of the revised manuscript.

2.2. Another aspect of the authors' analysis that raises concerns about the kinetochore-assembly model is their observation that cells with reduced SAM levels enrich in G1. One might reasonably expect kinetochore defective cells to arrest or delay in metaphase via the spindle assembly checkpoint, or perhaps slip out of mitosis with chromosome segregation errors increasing the frequency of senescence or apoptosis. The wording in the text suggests there's no increase in apoptosis, though I cannot tell if this was assayed in cells that were expected to be undergoing cell division. Is there evidence of aneuploidy (e.g. polyploidy/tetraploidy from failed mitoses; stain with centromeric markers and count pairs) or senescence in SAM deficient myoblasts?

Thanks for the excellent comment/suggestion. We agree the fate of the abnormally dividing cells needs to be investigated. Indeed, in many studies, kinetochore defective cells will show arrest or delay in metaphase via the spindle assembly checkpoint, or perhaps slip out of mitosis with chromosome segregation errors increasing the frequency of senescence or apoptosis^{6,7}. As suggested, we have examined for aneuploidy and found an increased number of aneuploidy cells in KO vs Ctrl (Fig. 6h). We have also repeated apoptosis assay in WT and KO satellite cells cultured for 2 days and confirmed no sign of apoptosis in KO cells. In addition, we have also performed SA- β -Gal staining and also found no sign of senescence in KO cells (Supplementary Fig. 6i). As answered above to the comments 2.1 and 1.3, we now believe the defect in kinetochore assembly upon *SAM* loss has led to extended G1 (Supplementary Fig. 1m) and premature differentiation (Fig. 4k, 4l and Supplementary Fig. 1n). In fact, Gogendeau, D et al. has described similar consequences in neural stem cells (NSCs) and intestine stem cells (ISCs) where they found that aneuploidy NSCs do not die by apoptosis. Instead, aneuploid NSCs display G1 lengthening and undergo premature differentiation³. Nonetheless, *Sugt1* depletion caused kinetochore defects and inhibited myoblast proliferation without leading to premature differentiation (Fig. 5h, 6a and Supplementary Fig. 5h). As answered to the above comment 2.1, we believe this discrepancy may be due to the degree of severity of kinetochore defects or the *Sugt1* independent function of *SAM* in myoblast differentiation. Ongoing efforts in the lab are directed toward a more detailed investigation of *Sugt1* function in SCs using genetic mouse model. The newly added results are presented in Fig. 6h, Supplementary Figs. 5h, 6i and Pages 16 and 19 on the revised text.

2.3. The current model of Sugt1/Sgt1 is that it acts as a co-chaperone for Hsp90, specifically linking Hsp90 to a subset of "clients". The data put forth by the authors suggests that SAM stabilizes Sugt1/Sgt1 and thus one might arguably stabilize the Hsp90-Sugt1/Sgt1 client or, as published previously, the Mis12 complex of the kinetochore. Given that there is specific client data, the authors should also measure the integrity of Mis12 complex under wild type and SAM KO conditions. The prediction is that it's the disruption of the Mis12 complex that ultimately impacts the quality of the Hec1/Ndc80 complex. This experiment is important as it connects the authors' model with known clients of the Hsp90-Sugt1/Sgt1 co-chaperone complex.

We thank the reviewer for the critical comment. We agree that further looking into the integrity of Mis12 complex will strengthen our mechanistic models. To this end, according to the suggestion, we have now examined the localization of one subunit of Mis12 complex, Dsn1 protein and found its enrichment was significantly decreased in the kinetochore regions in KO vs Ctrl cells of mitosis stage, confirming that *SAM* can facilitate the targeting of the Hsp90-Sugt1 "client" Mis12 complex to the

kinetochore; its loss leads to destabilization of Mis12 and subsequent degradation, which is consistent to what was observed upon Sugt1 depletion⁸. We also examined that localization of Hec1 and interestingly found an over-accumulation of this protein at the kinetochore region despite the unaltered total protein level. This is different from what was observed upon Sugt1 depletion in HeLa cells, which induced Hec1 destabilization and degradation. The reason behind the discrepancy needs to be further investigated. Nonetheless, previously Diaz-Rodriguez, E et. al.⁹ has shown that over-accumulation of Hec1 in mouse MEF cells can cause aberrant spindle phenotype. Altogether our findings thus suggest that SAM stabilizes Sugt1/Hsp90/Mis12 complex and facilitate proper Mis12 targeting to kinetochore, which is essential for the integrity of the entire kinetochore network. The newly added result is now in Fig. 6d and Page 18 on the revised text.

2.4. The other client based on the literature and highly related to the phenotype observed here, is Akt. Gao et al, (Mol Biol Rep) report that loss of Sg1 in cancer cells inhibits cell proliferation through dephosphorylation of Akt at position 473 (growth promoting). Deletion of SAM and destabilization of Sugt1 in myoblasts might be predicted to decrease Akt P-473 and thus inhibit myoblast proliferation. A Western blot using the phospho-specific antibody to Akt P473 would allow the authors to narrow the client range of Sugt1 in myoblasts or implicate a much more relevant pathway that would better explain the G1 enrichment in SAM knockouts.

Thanks for the excellent suggestion. As suggested, we have now examined the level of Akt ser473 phosphorylation and found it was in fact increased in KO vs Ctrl cells (Supplementary Fig. 6h), implying it may not be the Sugt1 client in myoblast cells thus may not be a relevant mediator of SAM/Sugt1 function in our cells. We have now added the above result on Page19 of the revised text.

2.5. The lack of developmental defects in animals with a SAM KO suggests that its role in stabilizing Sugt1 might be specific to myoblasts. This raises the possibility that Sugt1 is uniquely unstable in myoblasts compared to other tissues or in muscle cells during development. Minimally, it would be clarifying to test the Ub-modification or stability of Sugt1 in cells NOT involved in muscle regeneration in the SAM KO animals. This seems like a missing control to their Sugt1 levels experiment in Figure 5.

We thank the reviewer for the excellent comment. To include a control for the specific role of SAM in stabilizing Sugt1 in myoblasts, we have now managed to isolate primary hepatocytes from Ctrl and KO mice and found no decrease of Sugt1 level in KO vs Ctrl hepatocytes with or without CHX treatment, confirming the role of SAM in stabilizing Sugt1 might be specific to myoblasts. The newly added result can be found in Supplementary Fig. 5l and Page 17 on the revised text.

Other comments:

2.6. Sugt1/Sgt1 dependent assembly of kinetochore complexes was described in HeLa cells (Davies et al, JCB 2010). Inhibition of Sugt1/Sgt1 resulted in degradation of mis-assembled kinetochore complexes, a turnover that was dependent on Skp1-Ub ligases. After siRNA of both Sugt1/Sgt1 and Skp1, nonfunctional kinetochore complexes accumulated at centromeres, much as the authors see in myoblasts deficient in SAM. The authors' findings suggest that Ub-mediated degradation of kinetochore complexes might also be down-regulated in the SAM deficient cells, as mis-assembled kinetochore complexes appear to be associated with the centromeres. The authors should make this relationship clearer when pointing out that the decrease in Sugt1/Sgt1 doesn't match the published results. One might predict that the Skp1-Ub ligase complex is also destabilized by loss of SAM, or is inherently low in myoblasts. This is

first referred to on page 16 where the authors note the over-accumulation of Hec1/Ndc80, and then again in the Discussion on page 19.

We thank the reviewer for pointing out the interesting possibility. We have now performed the staining for Dsn1 in addition to Hec1. Interestingly, Dsn1 kinetochore signals were significantly decreased, in line with what was observed in Hela cells. According to Alexander ED et. al⁸, the degradation of Dsn1 is dependent on Skp-Ub ligase thus suggesting that this Ub pathway may be well functional in myoblast cells. Nevertheless, the kinetochore signals of Hec1 were increased and its total protein level remained unaltered, suggesting Hec1 may not be subject to Skp-Ub degradation in myoblasts. Altogether these findings reflect the similarity and difference between myoblast and Hela cells in terms of kinetochore assembly, which will be interesting to tease out in the future. As suggested, we have now made revisions on Pages 18 and 22 to discuss the point.

2.7. On page 16, the authors describe cold treatment of cells to measure Kt-Mt stability. Their sentence makes it seem as though SAM depleted cells had already been suggested to lack of stable kinetochore-microtubule attachments by Rieder, C. Et al. This sentence needs to be re-written to make it clear that loss of cold stable kt-mt attachments is an indication of kinetochore defects.

Thanks for your kind comment. We have now revised on Page 18 to clarify the point.

2.8. Page 3, second paragraph: “Cell cycle involves” should read “The cell cycle involves”. On the same line, “A faithful chromosome segregation”. On the next line, “In vertebrates, kinetochore” should read, “In vertebrates, the kinetochore”.

Thanks. We have now revised these places on Page 3.

2.9. Page 4, the authors spend some time discussing lncRNAs found associated with centromere function, but I’m not clear why this is relevant to their work. It makes the reader anticipate that SAM will be associated with centromeric heterochromatin, but this is not the case. I suggest eliminating or reducing this paragraph.

Thanks for the comment. We agree that the original writing on centromere lncRNAs may be too much, we have revised it on Page 5.

2.10. Page 7, second paragraph, the “pro-proliferating function of SAM” is awkward and inaccurate. The authors mean to say that SAM is required for efficient myoblast proliferation”. Page 11, first paragraph, “However, when examining more closely” should read, “However, when examined more closely”.

Thank! We have now revised these places on Pages 8 and 12.

2.11. Page 12, second paragraph, the comparison of MyoG+ and MyoD+ sentence is missing the KO vs. wild type text.

Thanks! We have now revised the writing on Page 14.

2.12. Overall the materials and methods, as well as the statistical analysis performed, are acceptable, and would allow a research with the proper expertise in muscle SC culturing to reproduce the experiments.

Reviewer #3:

This study reports the isolation and characterization of a lncRNA (SAM) expressed in activated satellite cells and proliferating myoblasts. Both cell culture experiments and generation of two independent SAM knock-out mice indicate a role for SAM in promoting cell proliferation and inhibiting myoblasts fusion and differentiation. Biochemical experiments permitted the identification of putative SAM-interacting proteins. Sug1 (suppressor of G2 allele of SKP1) was, among others, purified from C2C12 myoblasts lysates using biotin-labeled SAM transcripts. SAM-knock out cells displayed reduced Sug1 protein half-life, suggesting a role of SAM in regulating Sug1 protein stability. Sug1 knock-down experiments revealed defects in spindle formation and mitotic division. Similar defects were observed in SAM-knock out cells.

The phenotype observed in SAM-depleted cells and SAM-knock out mice are well documented. How SAM mechanistically regulates myogenesis remains to be firmly established. The SAM phenotypes are transient (3-4 days following injury) and not observed 1 month after injury.

We thank the reviewer for the positive comment and agree that additional work is needed to firmly establish the mechanism. Through answering all comments from the three reviewers/editors, we believe we now have more solid mechanistic data on how SAM regulates myogenesis.

3.1. Differences in WT and SAM KO cells are marginal, though statistically significant. Figure 4d reports a 6% (91.78% vs 85.61%) difference in Pax7+ MyoD+ cells in WT and SAM KO mice. Figure 4f indicates a 5.59% variation in Pax7+ MyoD + in WT and iKO cells.

Thanks for the comment. We agree the difference in Pax7+MyoD+ staining is not very high. This is because in Figs. 4d and 4f, cells were assayed at 48 hrs in culture when almost 90% cells had become fully proliferating myoblasts. However, when the assay was performed in an earlier stage before all cells became fully proliferating (20hr in culture), we observed 17.91% reduction in the proportion of Pax7 and MyoD double positive cells in KO vs. WT cells (Fig. 4J). In addition, EdU staining represents a much more sensitive method in detecting proliferation rate; in Figs. 4c and 4e, we observed a reduced percentage of EdU+ cell (14.8% and 25.61% respectively) in KO vs Ctrl cells. Nevertheless, we agree that the overall proliferative phenotype of KO cells is not dramatic considering the in vivo regenerative defect is relatively strong (Fig.2). We reason that the in vivo regenerative defect is a calmateive result from defects from multiple stages. As pointed out in the answer to Reviewer 1's comment 1.3, SAM loss may also lead to premature differentiation as well as other uncharacterized aspects. And these defects were further amplified in a chronic regenerative setting in SAM; mdx dKO mouse (Fig. 3). We have revised the text on Pages 22 and 23 to include the above points.

3.2. The RNA pull-down reported in Figure 5C and Supplemental Figure 7 document an apparent discrepancy in the Input lanes for Sug1 (obviously saturated) and those for the remaining proteins. It is

important for the authors to indicate the quantity of lysate (milligrams) employed in each experiment.

Thanks for your comment. We agree the input amount should be equal across all lanes. We have now replaced Fig. 5c with an image with equal amounts of input for all the examined proteins to show *SAM* retrieved Sugt1 but not hnRNPL, Dnmt3a and Dnmt3b, which are known lncRNA interacting partners. (Fig. 5c and raw images of the original blots in source data file). We also added the quantity of lysate used in the assay in the legend. To strengthen the data from full-length RNA pull down, we have now performed additional domain mapping assay to confirm the binding between *SAM* and Sugt1. However, we must point out that the retrieval efficiency in this assay appears to vary in between different batches of experiments although RIP and co-staining assays showed a more robust interaction. We reason it is possible that the interaction may depend on the modification for example the phosphorylation of Sugt1, which is only transiently present during prometaphase leading to the technical difficulty¹. It is also possible that RNA-protein pulldown assay is an in vitro assay which depends on the formation of RNA-protein complexes from synthetic target RNAs and cell extracts. Internal labeling of the target RNA with biotin has the potential downside of RNA misfolding due to steric hindrance. The newly added data can be found in Fig. 5c Supplementary Figs. 5a and b, as well as Page 16 and 42 on the revised text.

3.3. It is not clear how SAM prevents myoblast differentiation. Do Sugt1 knock-down cells display anticipated differentiation as SAM-knock out cells?

Thanks for your comment. We have answered this question to Reviewer 1' comment 1.3 and Reviewer 2's comments 2.1 and 2.2. Briefly, we believe that *SAM* loss causes kinetochore assembly defects thus delaying cell proliferation. Aneuploidy was observed which causes extended G1 and premature differentiation. Sugt1 knockdown has phenocopied the kinetochore defect and the delayed cell proliferation but not the premature differentiation. We speculate it is likely due to two reasons. First, Sugt1 KD may have caused a much more dramatic defects in kinetochore formation and cell proliferation, which may eventually lead to cell death thus prevention of differentiation; Second, *SAM* may have Sugt1 independent function in suppressing myoblast differentiation. Both possibilities can be investigated further in the next chapter of the study. For now, in the current manuscript we decide to focus on the proliferation aspect of *SAM* function in myoblasts. The newly added results can be found in Fig. 6f, 6g, 6h Supplementary Fig. 5h, 6f and 6g; as well as Pages 16 and 19 on the revised text.

3.4. Figure 2. eMyHC+ fibers decreased by 21.95% 4 days after injury. Was this difference observed at later times (14 days, 28 days following injury)?

Thanks for the suggestion. We have now performed eMyHC staining 7, 14 and 28 days following injury and found only subtle difference in KO vs Ctrl mice (Fig.2i). As expected, eMyHC was not found 28 days after injury as eMyHC is only expressed in regenerating but not mature fibers. The above results indicate that *SAM* deletion caused a delay but not blockage in acute injury induced muscle regeneration. Please also see answer to Reviewer 1's comment 1.3. This point is included in the revised text Page 11.

References

- 1 Liu, X. S. *et al.* Plk1 phosphorylates Sgk1 at the kinetochores to promote timely kinetochore-microtubule attachment. *Molecular and cellular biology* **32**, 4053-4067, doi:10.1128/MCB.00516-12 (2012).

- 2 Rocheteau, P., Gayraud-Morel, B., Siegl-Cachedenier, I., Blasco, M. A. & Tajbakhsh, S. A
subpopulation of adult skeletal muscle stem cells retains all template DNA strands after cell
division. *Cell* **148**, 112-125, doi:10.1016/j.cell.2011.11.049 (2012).
- 3 Gogendeau, D. *et al.* Aneuploidy causes premature differentiation of neural and intestinal stem
cells. *Nature communications* **6**, 8894, doi:10.1038/ncomms9894 (2015).
- 4 Carter, M. S. *et al.* A regulatory mechanism that detects premature nonsense codons in T-cell
receptor transcripts in vivo is reversed by protein synthesis inhibitors in vitro. *The Journal of
biological chemistry* **270**, 28995-29003, doi:10.1074/jbc.270.48.28995 (1995).
- 5 Kim, W., Kim, R., Park, G., Park, J. W. & Kim, J. E. Deficiency of H3K79 histone
methyltransferase Dot1-like protein (DOT1L) inhibits cell proliferation. *The Journal of
biological chemistry* **287**, 5588-5599, doi:10.1074/jbc.M111.328138 (2012).
- 6 Dobles, M., Liberal, V., Scott, M. L., Benezra, R. & Sorger, P. K. Chromosome
missegregation and apoptosis in mice lacking the mitotic checkpoint protein Mad2. *Cell* **101**,
635-645, doi:10.1016/s0092-8674(00)80875-2 (2000).
- 7 Baker, D. J. *et al.* BubR1 insufficiency causes early onset of aging-associated phenotypes
and infertility in mice. *Nature genetics* **36**, 744-749, doi:10.1038/ng1382 (2004).
- 8 Davies, A. E. & Kaplan, K. B. Hsp90-Sgt1 and Skp1 target human Mis12 complexes to ensure
efficient formation of kinetochore-microtubule binding sites. *The Journal of cell biology* **189**,
261-274, doi:10.1083/jcb.200910036 (2010).
- 9 Diaz-Rodriguez, E., Sotillo, R., Schvartzman, J. M. & Benezra, R. Hec1 overexpression
hyperactivates the mitotic checkpoint and induces tumor formation in vivo. *Proc Natl Acad
Sci U S A* **105**, 16719-16724, doi:10.1073/pnas.0803504105 (2008).

REVIEWERS' COMMENTS:

Reviewer #1 (Remarks to the Author):

The authors addressed all my concerns and I have no further comments.

Reviewer #2 (Remarks to the Author):

In the revised manuscript by Li et al, the authors make important strides in addressing the direct connection between SAM function, Sugt1 mediated kinetochore assembly and cell proliferation in skeletal muscle. Specifically, they have expanded their analysis to include siRNA of kinetochore proteins that demonstrate a delay in proliferation similar to that observed for SAM defects but without a premature differentiation. This confirms the notion that SAM defects separately impact kinetochores/myoblast proliferation and premature differentiation. This is clarifying though the authors speculate that SAM KO has a more dramatic kinetochore defect than a Sugt1 knockdown, which may explain the failure to observe premature differentiation. The authors go on to clarify the myoblast specific role of SAM and its impact on Sugt1, clarifying the narrowness of the developmental phenotype in their animals.

The authors also nicely addressed the idea that the Mis12 kinetochore complex is a likely client/target of Sugt1, further supporting their proposed connection between loss of SAM function and kinetochore assembly pathways. Their addition of data demonstrating the increase in Hec1 in the SAM knockouts nicely aligns their work with previous interrogation of Sugt1-Hsp90 kinetochore assembly pathways.

The only further issue to address for me is how the authors describe the connection between SAM, Sugt1, kinetochore assembly and skeletal muscle regeneration. In their rebuttal, the authors clearly state that the proliferation phenotype is linked to the Sugt1-dependent kinetochore assembly defect, but not as clearly to myogenic differentiation. I can appreciate that additional roles of SAM in differentiation will require separate work, but I also maintain that their title and abstract are a bit misleading on this front. The authors need to state this distinction more clearly, at least in their title and abstract. Their argument that other systems link aneuploidy to differentiation may or may not be relevant here, but are appropriate for their Discussion. In the end, the authors have nicely demonstrated that SAM regulates cell proliferation through its impact on Sugt1 and kinetochore assembly and it may have additional roles in myoblast differentiation.

I recommend publication after addressing this wording issue.

Reviewer #3 (Remarks to the Author):

The authors have addressed several points raised by the reviewers.

SAM KO mice displayed regenerative defects limited to 3-4 days after injury and recovered starting from day 7 with fully regenerated fibers by day 28 (Figure 2).

SCs isolated from SAM KO mice displayed evident defects in chromosome alignment, mitotic spindle formation (Figure 6), increased microtubules instability, altered localization of kinetochore components, and aneuploidy. However, neither apoptosis nor senescence was increased in SAM KO SCs (Supplementary Figure 4f and Supplementary Figure 6i). One is left wondering how SCs so deranged in their cellular organization, and not selected against by either apoptosis or senescence, go on to regenerate normal muscle.

Reviewer #2:

In the revised manuscript by Li et al, the authors make important strides in addressing the direct connection between SAM function, Sugt1 mediated kinetochore assembly and cell proliferation in skeletal muscle. Specifically, they have expanded their analysis to include siRNA of kinetochore proteins that demonstrate a delay in proliferation similar to that observed for SAM defects but without a premature differentiation. This confirms the notion that SAM defects separately impact kinetochores/myoblast proliferation and premature differentiation. This is clarifying though the authors speculate that SAM KO has a more dramatic kinetochore defect than a Sugt1 knockdown, which may explain the failure to observe premature differentiation. The authors go on to clarify the myoblast specific role of SAM and its impact on Sugt1, clarifying the narrowness of the developmental phenotype in their animals.

The authors also nicely addressed the idea that the Mis12 kinetochore complex is a likely client/target of Sugt1, further supporting their proposed connection between loss of SAM function and kinetochore assembly pathways. Their addition of data demonstrating the increase in Hec1 in the SAM knockouts nicely aligns their work with previous interrogation of Sugt1-Hsp90 kinetochore assembly pathways.

The only further issue to address for me is how the authors describe the connection between SAM, Sugt1, kinetochore assembly and skeletal muscle regeneration. In their rebuttal, the authors clearly state that the proliferation phenotype is linked to the Sugt1-dependent kinetochore assembly defect, but not as clearly to myogenic differentiation. I can appreciate

that additional roles of SAM in differentiation will require separate work, but I also maintain that their title and abstract are a bit misleading on this front. The authors need to state this distinction more clearly, at least in their title and abstract. Their argument that other systems link aneuploidy to differentiation may or may not be relevant here, but are appropriate for their Discussion. In the end, the authors have nicely demonstrated that SAM regulates cell proliferation through its impact on Sugt1 and kinetochore assembly and it may have additional roles in myoblast differentiation.

I recommend publication after addressing this wording issue.

Thanks for the comment and we have addressed this wording issue.

Reviewer #3:

The authors have addressed several points raised by the reviewers.

SAM KO mice displayed regenerative defects limited to 3-4 days after injury and recovered starting from day 7 with fully regenerated fibers by day 28 (Figure 2).

SCs isolated from SAM KO mice displayed evident defects in chromosome alignment, mitotic spindle formation (Figure 6), increased microtubules instability, altered localization of kinetochore components, and aneuploidy. However, neither apoptosis nor senescence was increased in SAM KO SCs (Supplementary Figure 4f and Supplementary Figure 6i). One is left wondering how SCs so deranged in their cellular organization, and not selected against by either apoptosis or senescence, go on to regenerate normal muscle.

We thank the reviewer for the critical comment. We have addressed this concern in the Discussion of the manuscript on Page 22. Briefly, we believe that despite the evident defects in mitotic division, the cells did not undergo apoptosis or senescence, instead, premature differentiation occurs, leading to relatively mild regenerative phenotype in one- round of injury/regeneration. However, the phenotype was much more pronounced in dKO that underwent chronic regeneration allowing the defects in SCs to be amplified.